# Prolonged hospitalization signature and early antibiotic effects on the nasopharyngeal resistome in preterm infants

Achal Dhariwal [●][1,8], Polona Rajar[1,2,8], Gabriela Salvadori[1], Heidi Aarø Åmdal[1], Dag Berild[3,4], Ola Didrik Saugstad[5], Drude Fugelseth[2,4], Gorm Greisen [●][6], Ulf Dahle [●][7], Kirsti Haaland[2] & Fernanda Cristina Petersen [●][1] ✉

Respiratory pathogens, commonly colonizing nasopharynx, are among the leading causes of death due to antimicrobial resistance. Yet, antibiotic resistance determinants within nasopharyngeal microbial communities remain poorly understood. In this prospective cohort study, we investigate the nasopharynx resistome development in preterm infants, assess early antibiotic impact on its trajectory, and explore its association with clinical covariates using shotgun metagenomics. Our findings reveal widespread nasopharyngeal carriage of antibiotic resistance genes (ARGs) with resistomes undergoing transient changes, including increased ARG diversity, abundance, and composition alterations due to early antibiotic exposure. ARGs associated with the critical nosocomial pathogen *Serratia marcescens* persist up to 8–10 months of age, representing a long-lasting hospitalization signature. The nasopharyngeal resistome strongly correlates with microbiome composition, with inter-individual differences and postnatal age explaining most of the variation. Our report on the collateral effects of antibiotics and prolonged hospitalization underscores the urgency of further studies focused on this relatively unexplored reservoir of pathogens and ARGs.

Antimicrobial-resistant infections are a major global health threat. Among these, respiratory infections account for more than 1.5 million deaths a year, surpassing the number of deaths caused by resistant infections in any other part of the human body[1]. Main pathogens include *Streptococcus pneumoniae, Staphylococcus aureus, Klebsiella pneumoniae*, and *Haemophilus influenzae*, all generally present in the nasopharynx of healthy populations. These pathogens share common niches with commensals, which in turn provide colonization resistance against pathogens. However, commensals can also serve as reservoirs of antibiotic resistance genes (ARGs) that can readily be transferred to pathogens[2–4].

Recent studies have highlighted the potential negative impact of antibiotic treatment on the nasopharynx microbiome and its association with an increased risk of developing asthma and respiratory infections in infants[5–7]. These studies have significantly advanced our understanding of the ecological impact of antibiotics on respiratory health and disease. However, as these studies rely on 16S rRNA sequencing technologies, they do not offer a high level of microbial taxonomic resolution nor provide information on the overall gene composition, including the presence of ARGs within microbial communities, referred to as the resistome. In our previous study, we developed an optimization strategy for DNA extraction and library

[1]Institute of Oral Biology, Faculty of Dentistry, University of Oslo, Oslo, Norway. [2]Department of Neonatal Intensive Care, Division of Pediatric and Adolescent Medicine, Oslo University Hospital, Oslo, Norway. [3]Department of Infectious Diseases, Oslo University Hospital, Oslo, Norway. [4]Institute of Clinical Medicine, Faculty of Medicine, University of Oslo, Oslo, Norway. [5]Department of Pediatric Research, University of Oslo, Oslo, Norway. [6]Department of Neonatology, Copenhagen University Hospital Rigshospitalet, Copenhagen, Denmark. [7]Centre for Antimicrobial Resistance, Norwegian Institute of Public Health, Oslo, Norway. [8]These authors contributed equally: Achal Dhariwal, Polona Rajar. ✉e-mail: f.c.petersen@odont.uio.no

preparation that helped to circumvent the challenges posed by low bacterial biomass and high host DNA content in nasopharynx samples from preterm infants[8]. In the present study, we applied this optimized protocol to nasopharyngeal aspirate samples to characterize the resistome development in 36 preterm infants from birth to 6 months corrected age (8–10 months chronological age). By analyzing ~1.85 terabytes (Tb) of metagenomic DNA data generated from deep shot-gun metagenomic sequencing, we obtained the most comprehensive view to date of the characteristics and dynamics of resistome development in the respiratory tract of preterm infants. Our hypothesis was that early exposure to broad-spectrum antibiotics for suspected early onset sepsis (sEONS) treatment in preterm infants may lead to prolonged compositional alterations in the respiratory resistome. We also explored the extensive metadata collected to gain insights into additional clinical covariates that may influence resistome development.

## Results

### Distribution and characterization of ARGs in the preterm infant nasopharyngeal microbiome

In total, we found 369 ARGs belonging to 15 ARG classes conferring resistance through 5 distinct mechanisms. ARGs associated with multidrug efflux pumps comprised the majority of the resistome with an average of 27% from T1 to T5 and 17.8% at T6 (six months corrected age) of the total relative abundance. There was a mean of 24 (median: 18) unique ARGs ranging from 1 to 155 per analyzed sample. On average, the maximum number of ARGs were detected after the discontinuation of treatment (T2) (mean: 40) in most infants receiving early antibiotic treatment. In the naive group, the corresponding mean value at T2 was 19. Only 13 ARGs were found to be both highly prevalent (detected in >40% of all samples) and highly abundant, with a mean relative abundance of 65.2% (SD: 32%) across all preterm infant nasopharyngeal resistome. These core set of resistance genes included multidrug resistance genes (*acrB, oqxB, mexI*), macrolide-lincosamide-streptogramin (MLS) resistance genes (*mel, pmrA, rlmA(II)*), two beta-lactam resistance genes (*blaZ, SST-1*), fluoroquinolone resistance genes (*patA, patB*), tetracycline resistance genes (*tet(41), tetM*), and the aminoglycoside resistance gene *AAC(6')-Ic*.

In both antibiotic-treated and naive groups, the most abundant class of ARG was multidrug (mean: 27.95%; SD: 21.96%), followed by fluoroquinolone (mean: 17.45%; SD: 20.28%), beta-lactam (mean: 15.35%; SD: 17.05%), tetracycline (mean: 14.33%; SD: 10.93%), MLS (mean: 13.18%; SD: 15.43%) and aminoglycoside (mean: 8.95%; SD: 12.94%) (Fig. 1A). In terms of prevalence, aminoglycoside (76/82), and beta-lactam (75/82) were the most common classes present in the samples from antibiotic-treated group, while multidrug (92/99) and beta-lactam (91/99) were the most common classes in the naive infants. When classified based on resistance mechanism, antibiotic efflux (mean: 59.17%; SD: 21.54%) was the most abundant category in both the groups across all time points (Supplementary Fig. 4). At baseline (T1), the ARG composition of the antibiotic-treated group was found to be more heterogeneous (higher inter-individual variability) and diverse compared to that of naive infants (Fig. 1B and Supplementary Fig. 10).

The health risks of ARGs were assessed using the recently proposed ARG-rank list, in which Rank IV comprises ARGs not associated with humans, Rank III includes human-associated ARGs found in non-mobile elements, Rank II encompasses mobile ARGs capable of transferring existing resistance to pathogens, and Rank I comprises ARGs already present in pathogens, thus posing the highest risk to human health[9]. We detected homologs to potential high-risk ARGs (rank I & II) in the nasopharyngeal samples at multiple time points across infants in both antibiotic-treated and naive groups (Supplementary Fig. 5). We also observed a few high-risk and clinically relevant ARGs that appeared directly after the early antibiotic treatment at T2. For instance, high-risk ARGs specific to the antibiotics encoding for

aminoglycoside-modifying enzymes (*AAC(3)-II*) along with extended-spectrum TEM beta-lactamase (*bla_{TEM-1}*) were detected in two infants. These ARGs co-occurred with other non-targeted high-risk (rank II) ARGs such as *qnrS, floR*, and *aadA*. Previous studies have shown the co-carriage of these ARGs on multidrug resistance plasmids in Enterobacter species[10]. Also, other clinically relevant Extended Spectrum Beta-Lactamase (ESBL) genes belonging to the SHV beta-lactamase (rank II) family were detected in two other infants following antibiotic exposure. In one patient, the plasmid-mediated AmpC-type β-lactamase (ACT-beta lactamase), known to confer resistance against all classes of beta-lactam antibiotics such as penicillins, cephalosporins, and carbapenems was detected only after the treatment cessation (T2). These ARGs were mainly observed in early antibiotics exposed infants whose mothers had also received antibiotics during pregnancy (prenatal). Nonetheless, they did not persist in nasopharyngeal samples collected after discharge at 6 months corrected age (T6). Amongst ARGs found in Gram-positive pathogens, the *Staphylococcal* methicillin-resistance gene i.e., *mecA* was detected in 44 samples from almost all (14 of 15) antibiotic-treated infants and 10 of the 21 naive infants.

### Impact of early antibiotics on preterm infant nasopharyngeal resistome

Firstly, we investigated the total ARG abundance (quantified as the sum of RPKM) and ARG α-diversity (quantified by the Shannon index) across sampling time points. In both the antibiotic-naive and treated groups, we found that total ARG abundance and diversity increased from T1 to T3-T4, before descending at T5 to the 6 months corrected age (T6). In treated infants, total ARG abundance and diversity increased more rapidly upon antibiotic treatment, whereas the changes were generally more gradual over time points in naive infants (Fig. 2A, B). To identify the direct effects of early antibiotic treatment on the total ARG abundance and diversity, we used a generalized linear mixed model with individual set as random effect while correcting for age (DOL). Early antibiotics were significantly associated with increase in both total ARG abundance (LME model: *p.adj* = 0.01) and Shannon diversity (*p.adj* = 0.01) after treatment (T2) as compared to the baseline (T1) samples in preterm infant nasopharyngeal resistome. Moreover, the significant effect of antibiotics on the increase in total ARG abundance was observed until T3 (*p.adj* = 0.001). Nonetheless, such effects were short-lived, as we did not identify significant differences at any of the following time points compared to the baseline (T1) in antibiotic-treated group. In the naive group of infants, no significant differences appeared in total ARG abundance and α-diversity between any time points during the first 6 months (corrected) of life while correcting for age.

β-diversity analysis revealed that the overall resistome composition did significantly differ between the two groups (PERMANOVA: $R^2$ = 8.4%, *p.adj* = 0.02) at the baseline (T1) (Fig. 3A). Infants that received early antibiotic treatment had more dissimilar (dispersed) resistome composition than the naive infants (Wilcoxon test: *p* = 2.2e−05; Supplementary Fig. 10A). Further, we observed that nasopharyngeal samples were clustering more effectively according to birth weight group ($R^2$ = 16.5%, *p.adj* = 0.01; Supplementary Fig. 6A). No significant difference in the baseline resistome composition between antibiotic-treated and naive infants ($R^2$ = 3.9%, *p.adj* = 0.26) was found while correcting for birth weight group. However, a minor yet significant compositional difference in the resistome was still observed between the two groups ($R^2$ = 6.4%, *p.adj* = 0.02; adjusted for BW group: $R^2$ = 5.9%, *p.adj* = 0.03) directly after the cessation of early antibiotic treatment (T2) (Fig. 3B). Moreover, the compositional changes due to early antibiotics at T2 were more discernible in infants whose mothers had received prenatal antibiotics during pregnancy (prenatal + early (*n* = 8) vs naive (*n* = 20): $R^2$ = 10.1%,

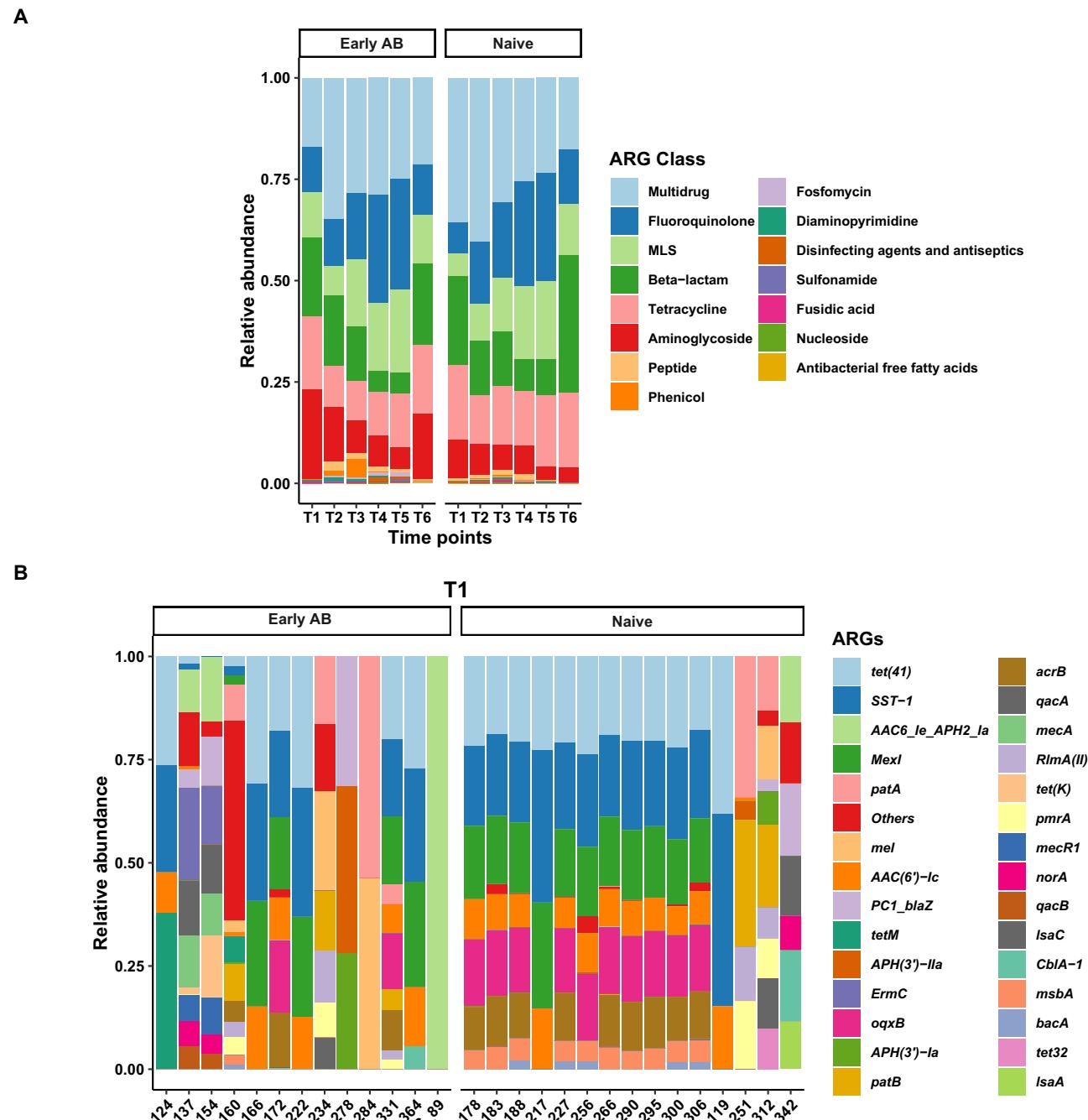

**Fig. 1 | Resistome composition in the nasopharynx microbiome of preterm infants.** Stacked bar plots showing the relative abundance of **A** antibiotic resistance genes (ARGs) clustered at the Class level for the naive and early antibiotic-treated groups over time; and **B** the top 30 most abundant ARGs identified across samples at baseline (T1) in antibiotic-treated (left) and naive (right) groups. All other ARGs are grouped into the "Others" category. MLS macrolide-lincosamide-streptogramin. Source data are provided as a Source data file.

$p.adj = 0.009$; Supplementary Fig. 6B). Following treatment, the resistome normalized rapidly as no significant differences in composition between the naive and treated groups were detected from T3 until 6 months of corrected age (T6) (Fig. 3C).

### Clinical covariates associated with the development of preterm infant nasopharyngeal resistome composition

The associations between resistome composition and clinical factors were analyzed using PERMANOVA-tests (all $p.adj \leq 0.001$) and principal component analysis (PCA) across all nasopharyngeal samples from preterm birth until 6 months corrected age, using centered log-ratio (CLR) transformed data (Supplementary Fig. 2). The PCA was explained by two main principal components, describing 18.7% and 17.6% of the variation, respectively. Besides inter-individual variation (PERMANOVA (univariate): $R^2 = 37.8\%$, $p.adj = 0.001$), age was the other significant covariate found to be associated with overall resistome composition in the overarching cohort. A moderate yet statistically significant effect was observed with GA ($R^2 = 4\%$, $p.adj = 0.001$) and BW group ($R^2 = 4.1\%$, $p.adj = 0.001$), which are highly correlated variables (as indicated by

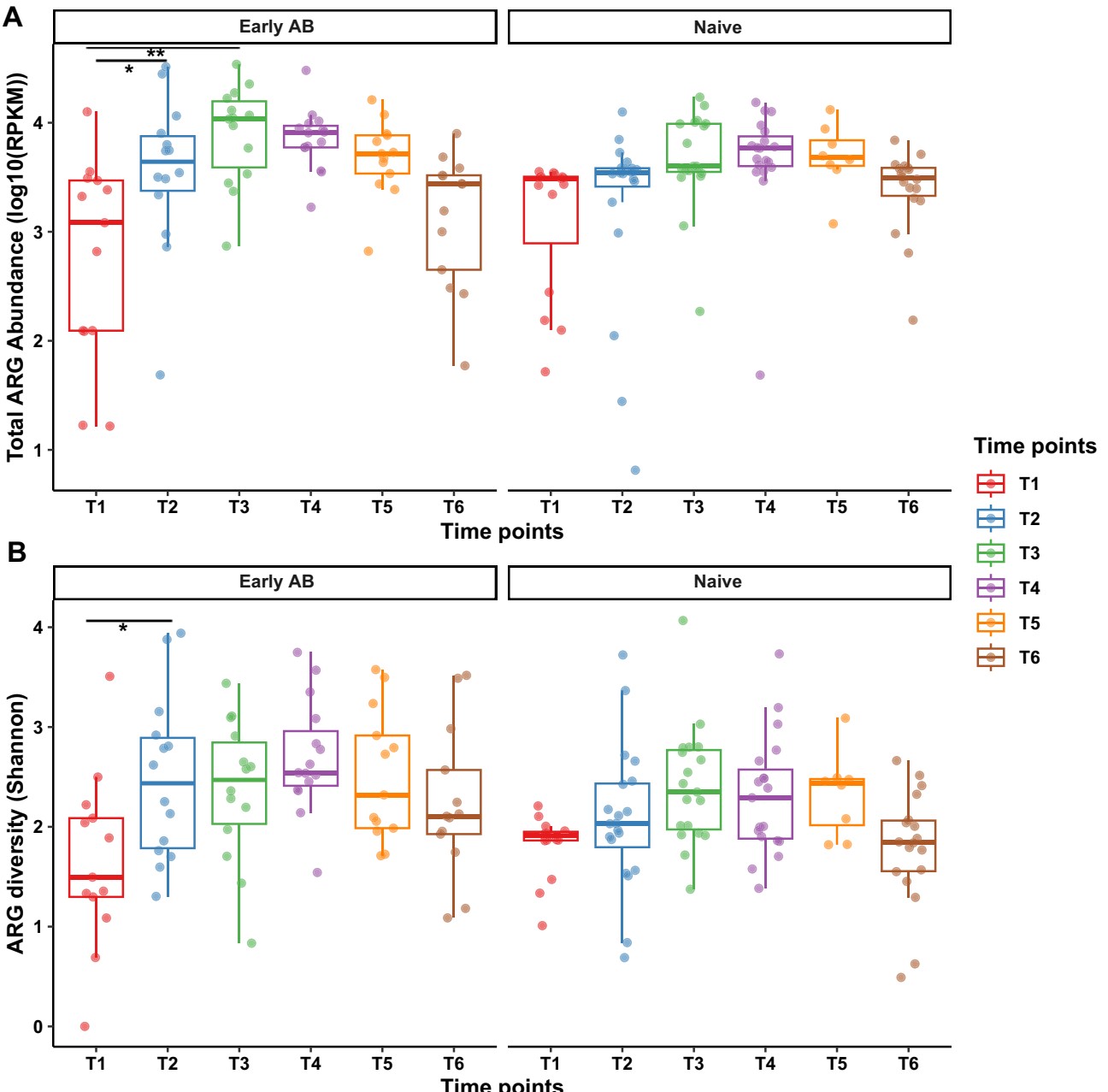

**Fig. 2 | Impact of early antibiotic treatment on nasopharynx total ARG abundance and ARG α-diversity.** Boxplots showing the **A** total ARG abundance (sum of RPKM) and **B** ARG Shannon diversity per sample, stratified by antibiotic treatment and time points. The lower and upper horizontal box lines correspond to the first and third quartiles, with the line in the middle of the boxes representing the median. Whiskers extending from the boxes represent the range of data points within 1.5 times the interquartile range below the first quartile and within 1.5 times the interquartile range above the third quartile. All data points are visualized using translucent circles: red = T1 (*n* = 13 for Early AB and *n* = 15 for Naive), blue = T2 (*n* = 14 for Early AB and *n* = 20 for Naive), green = T3 (*n* = 14 for Early AB and *n* = 19 for Naive), purple = T4 (*n* = 15 for Early AB and *n* = 19 for Naive), orange = T5 (*n* = 13 for Early AB and *n* = 8 for Naive), and brown = T6 (*n* = 13 for Early AB and *n* = 18 for Naive). Significant differences over time points in the treatment groups were calculated using an LME model with subject as random effect and two-sided Tukey's HSD post hoc tests. Adjusted *P*-values were generated by the Benjamini–Hochberg correction procedure. We observed a significant difference in the total ARG abundance (*p.adj* = 0.01) and ARG Shannon diversity (*p.adj* = 0.01) in the antibiotic-treated group (Early AB) directly after the antibiotic treatment (T2) compared to the baseline samples (T1), as indicated by asterisks. A significant increase in the total ARG abundance was also observed at T3 compared to T1 in antibiotic-treated group (*p.adj* = 0.001). \**p* < 0.05, \*\**p* < 0.01 and \*\*\**p* < 0.001. LME linear mixed effect, *p.adj* adjusted *P*-values, RPKM reads per kilobase of reference gene per million bacterial reads. Source data are provided as a Source data file.

the chi-square test: *p* < 2.2e−16). Mode of delivery, sex, and intra-partum antibiotic treatment did not appear to shape the overall nasopharyngeal resistome composition (Supplementary Data 2). Covariates that differed between the antibiotic-treated and naive infants at baseline and that also had a significant association with the overall resistome composition (GA and prenatal antibiotics (R² = 4.5%, *p.adj* = 0.001)) were used for correction in the subsequent multivariate analysis.

Multivariate PERMANOVA analysis showed that the largest effect on overall resistome composition remained attributed to inter-individual variation (R² = 30.13%, *p.adj* = 0.001), followed by PMA (R² = 9.8%, *p.adj* = 0.001) and the chronological age group (sampling

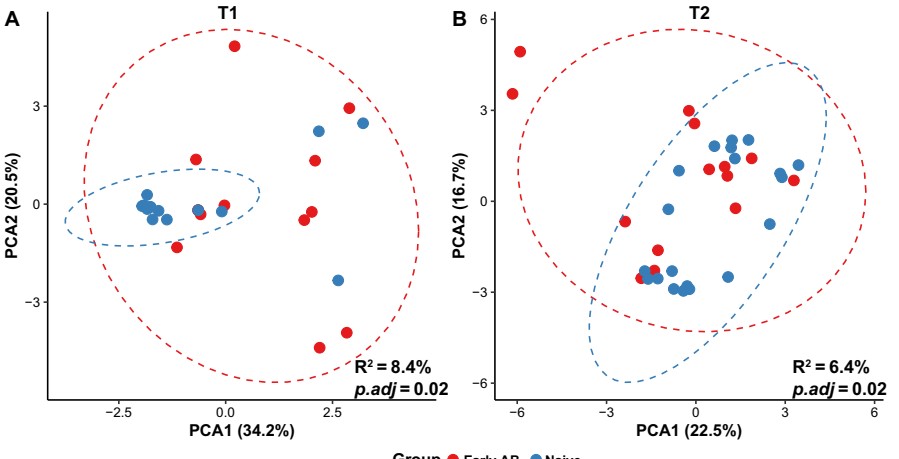

| Time points | R² (%) | p.adj |
|---|---|---|
| T3 | 3.4% | 0.345 |
| T4 | 4.4% | 0.148 |
| T5 | 5.1% | 0.371 |
| T6 | 3.6% | 0.355 |

**Fig. 3 | β-diversity PCA plots of overall nasopharyngeal resistome composition stratified per time point.** Principal component analysis (PCA) based on Euclidean distances of CLR-transformed ARG abundances (i.e., Aitchison dissimilarity) between samples, visualizing the overall nasopharyngeal resistome composition stratified for antibiotic-treated infants and naive at time point **A** T1 (baseline) and **B** T2 (immediately after cessation of early antibiotic therapy). Each data point represents the resistome composition of one sample. The ellipses represent the 95% confidence interval around the centroid of resistome composition for each group. Statistical significance (adjusted *P*-values, calculated using the Benjamini–Hochberg method) and effect size ($R^2$) of the differences in beta diversity were assessed using PERMANOVA test. In panel (**C**), the results of the PERMANOVA-tests for the later time points are presented. Data points of early antibiotic-treated infants are colored red and those of naive infants are blue. PERMANOVA permutational multivariate analysis of variance; *p.adj* = adjusted *P*-values. Source data are provided as a Source data file.

time points) ($R^2 = 8.4\%$, *p.adj* = 0.001), despite a high degree of variation between samples with no obvious discrete spatial clustering in relation to these variables upon visual inspection (Fig. 4A–C). We found no significant influence of early antibiotics ($R^2 = 0.6\%$, *p.adj* = 0.339) and its duration ($R^2 = 1.3\%$, *p.adj* = 0.007) after adjusting for confounding variables. However, we observed a significant effect of prenatal antibiotics ($R^2 = 3.6\%$, *p.adj* = 0.001) and its duration (categorized) ($R^2 = 4.4\%$, *p.adj* = 0.001) in shaping the overall nasopharyngeal resistome composition in preterm birth. To account for the likelihood of prenatal antibiotics with early antibiotics exposure in our analyses (Chi-squared test, *p* < 0.0003), we further categorized our early antibiotic-treated group based on whether the mothers had also received antibiotics during pregnancy (prenatal + early, *n* = 9) or not (only early, *n* = 6). Post hoc analysis revealed significant differences in the overall resistome composition of infants having exposure to both prenatal and early life antibiotics (prenatal + early) compared to infants receiving either only early or naive (no prenatal + no early) (*p.adj* = 0.001; Fig. 4D). No difference in overall resistome composition was observed between the only early antibiotics and naive groups (*p.adj* = 0.06), underlining the relevance of prenatal antibiotic exposure in shaping the overall resistome composition of preterm infants during the first 6 months (corrected) of life.

### Unsupervised clustering classifies nasopharyngeal samples into distinct clusters based on composition differences

Next, we determined whether the nasopharyngeal samples were organized into clusters (i.e., resistotypes) according to their ARG abundance profiles using an unsupervised method, i.e., DMM models. This way, we identified three distinct clusters (or resistotypes) that were driven by specific ARGs. The majority of our samples (~87%) were classified into two main resistotypes, R1 and R2. The first resistotype (R1) was characterized by a predominance of fluoroquinolone resistance genes (*patA, patB*) and *rlmA(II)* (Supplementary Fig. 7). On the other hand, resistotype R2 was driven mainly by relatively higher abundance of the beta-lactam resistance gene *SST-1*, aminoglycoside resistance gene *AAC(6')-Ic*, tetracycline resistance gene *tet(41)* and the multidrug resistance gene *mexI*. Resistotype R3 was enriched with the *blaZ* β-lactamase resistance gene. The identified resistotypes varied in

their overall ARG composition (β-diversity), as shown in a PCA plot (Fig. 5A). Furthermore, PERMANOVA analysis showed that resistotypes were more strongly (larger effect size) and significantly associated with the resistome than other clinical covariates or early-life factors in preterm infants during the first 6 months (corrected) of life ($R^2 = 20.4\%$, *p.adj* = 0.001).

### Nasopharyngeal resistome composition of preterm infants is linked to microbial composition and carries hospitalization signatures

To investigate the impact of microbial composition on the resistome composition, we taxonomically profiled the infant nasopharyngeal samples using MetaPhlAn3[11] (Supplementary Fig. 9). We found that microbial community types, as determined by DMM models, were highly linked to the resistotypes (chi-square test: *p* < 0.0001). Resistotype R2 was very specific to the *Serratia* community type. Resistotype R1 was associated with community types mainly driven by *Streptococcus* and *Gemella*, while Resistotype R3 was enriched in community types predominated by *Staphylococcus* and *Streptococcus* (Fig. 5B). These community types were also significantly associated with the overall composition of the resistome (PERMANOVA: $R^2 = 13.5\%$, *p.adj* = 0.001; Fig. 5C). In addition, results from Procrustes analysis showed a strong and significant association between the microbial taxa and ARG abundance profiles (PROTEST: correlation coefficient (r) = 0.87, *p* = 0.001, sum of squares ($m^2$) = 0.24), underlining that bacterial community composition structured the resistome composition in the nasopharyngeal microbiome of preterm infants. However, the correlation was more robust in the antibiotic-naive infants (PROTEST: r = 0.94, *p* = 0.001, sum of squares ($m^2$) = 0.12) as compared to antibiotic-treated infants (PROTEST: r = 0.85, *p* = 0.001, sum of squares ($m^2$) = 0.27; Fig. 6A, B).

Next, we performed pairwise Spearman's correlation analysis between ARG and microbial species abundances to predict the origin or potential microbial host of ARGs in nasopharyngeal samples. The strongest positive correlation (r ≥ 0.8, *p.adj* < 0.05) was found between the *SST-1, AAC(6')-Ic, tet(41)* ARGs, and *Serratia marcescens/nematodiphila* (Fig. 6C). It is worth noting that during our study, a clinically detected outbreak of *S. marcescens* occurred in the NICU a few months

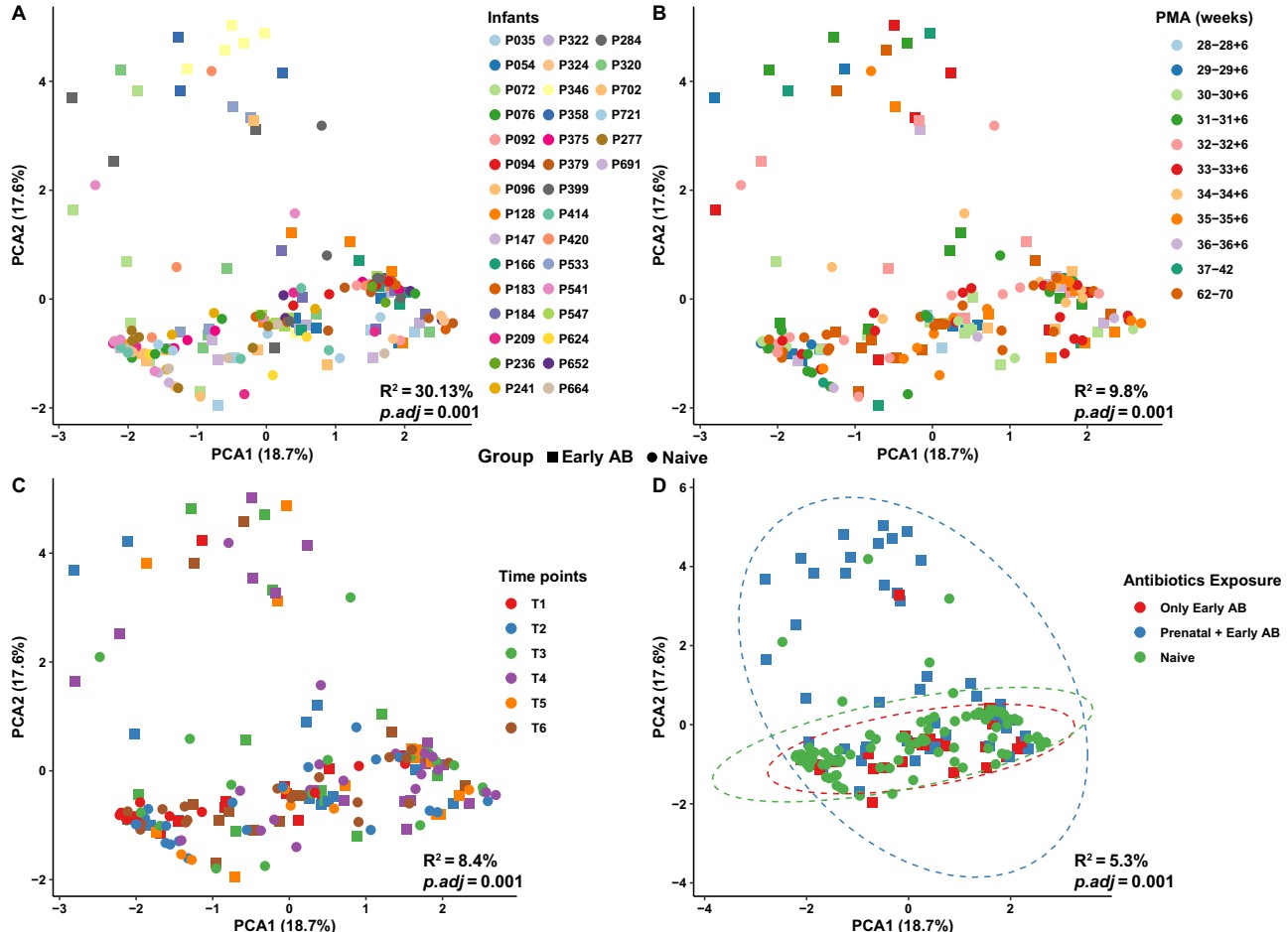

**Fig. 4 | Factors influencing the overall nasopharyngeal resistome composition in the preterm infants.** PCA visualizations of Aitchison dissimilarities between all the nasopharyngeal samples from birth until 6 months corrected age at ARG level. Each data point corresponds to a sample, colored by **A** Infant, **B** PMA, **C** Time points, and **D** Prenatal and post-natal antibiotic exposure and shaped according to early antibiotic exposure or naive groups. Effect sizes ($R^2$) and corresponding adjusted $P$-values (*p.adj*, corrected using the Benjamini–Hochberg method) are calculated using PERMANOVA-tests (as shown in the bottom right of each plot). Percentage refers to the percentage of variance explained by the principal component. PMA postmenstrual age, PCA principal component analysis, PERMANOVA permutational multivariate analysis of variance, *p.adj* adjusted $P$-values. Source data are provided as a Source data file.

after infant enrollment had commenced. When present at T1, the carriage of these ARGs persisted in 24 out of 26 preterm infants across both naive and treated groups, extending long after hospital discharge as observed at ~8 to 10 months chronological age (i.e., 6 months corrected age (T6)) (Fig. 6D). Occasional time points in which ARGs associated with *S. marcescens* showed variations from T1 were observed in few of the samples. Whether these are true variations is difficult to ascertain as metagenomic studies can only detect DNA sequences that are present in specific samples and to a depth that is possible to be captured by present analytical approaches.

The highly prevalent ARGs like *patA, patB* and *rlmA(II)* within Resistotype 1 were observed to exhibit a strong co-occurrence with *Streptococcus mitis/oralis* and *Gemella haemolysans/sanguinis* (Fig. 6C). *Staphylococcus aureus* and *Staphylococcus epidermidis* correlated only with *blaZ* beta-lactam ARG. These associations are in accordance with information in the literature and AMR gene databases on the known microbial hosts of these ARGs, highlighting the potential of such an analytical approach to predict the origin of ARG in metagenomes. However, to reduce the bias due to false ranking of features with many zero values, this analysis is only applicable on the features (ARGs or taxa) present in high prevalence across samples.

Lastly, we also examined whether some of these strongly associated bacterial species with ARGs would also directly correlate with our overall resistome outcomes. We plotted the relative abundance of each species versus the total ARG abundance (sum of RPKM) and the number of unique ARGs from all samples. Our analysis revealed that only the relative abundance of *Streptococcus mitis* (rmcorr correlation coefficient ($r^{rm} = 0.43$, *p.adj* = 5.82e−08) showed a moderately linear association with the total ARG abundance while adjusting for inter-individual variability (Supplementary Fig. 8). In addition, we observed a small but significant association between the relative abundance of *S. mitis* and the number of unique ARGs ($r^{rm}$ = 0.19, *p.adj* = 0.02). To statistically assess whether *S. mitis* relative abundance explains the dispersion of resistomes across samples, we conducted the PERMANOVA test and found that *S. mitis* significantly explains 8.5% (*p.adj* = 0.001) of the variation in the overall resistome composition. Only *S. marcescens* ($R^2$ = 13.9%, *p.adj* = 0.001) accounts for a higher proportion of the total variance in the resistome composition.

## Discussion

The nasopharynx represents a highly accessible microbial community and also serves as a crucial diagnostic window in combating respiratory infections and AMR[12]. It is also the primary ecological niche for common respiratory pathogens[13]. To date, one of the closest attempts at an in-depth investigation of ARGs in the respiratory tract using a culture-free approach identified ARGs in 86% of the samples. However,

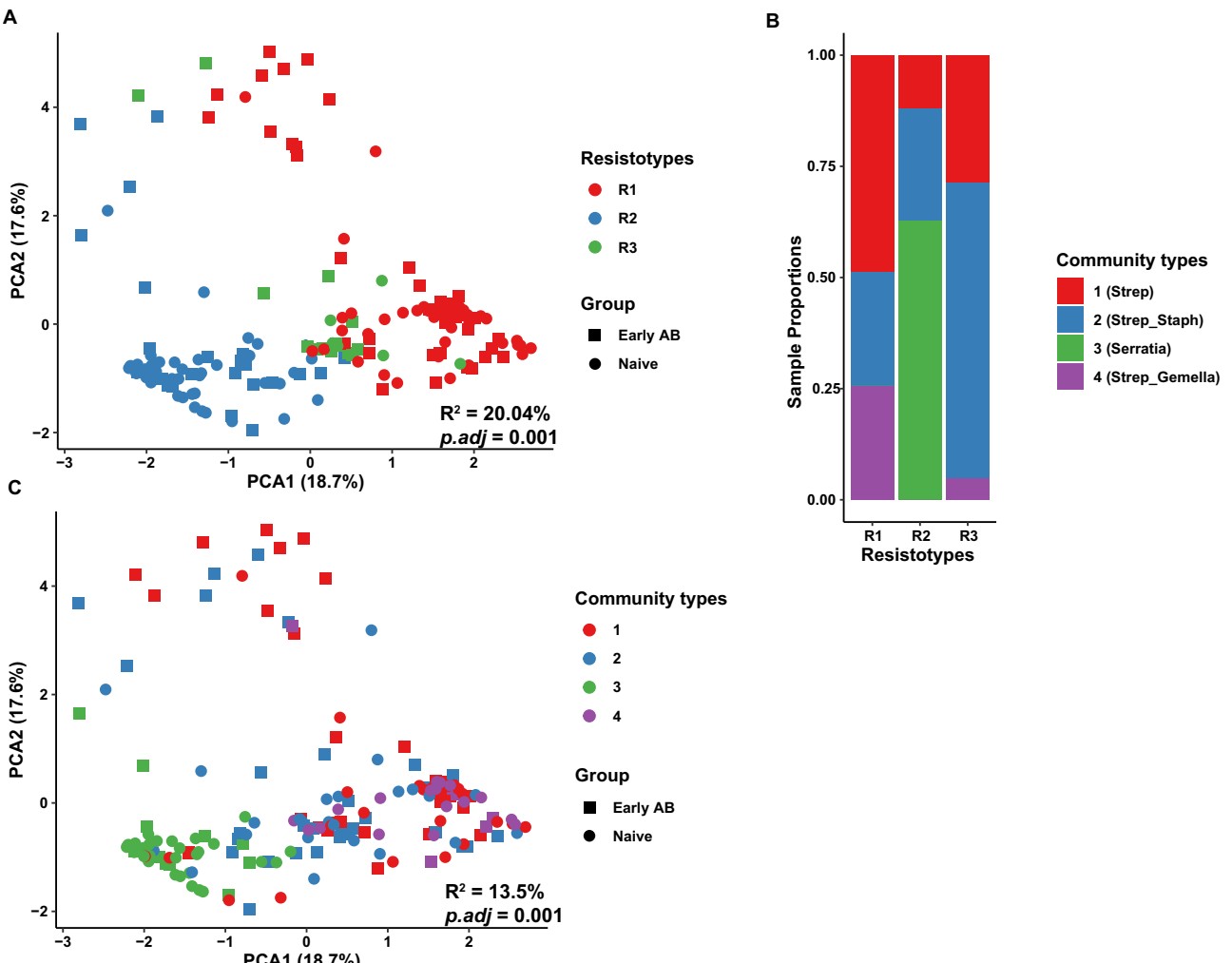

**Fig. 5 | Association between resistotypes, community types, and ARG abundance profiles.** The PCA plot showing the association between overall ARG abundance profile with respect to their **A** resistotypes and **C** community types identified among all the nasopharyngeal samples using the DMM models. $R^2$ and adjusted *P*-values (*p.adj*, adjusted using the Benjamini–Hochberg method) was obtained using the PERMANOVA test. **B** Samples proportions for each resistotype shown as a function of community type. DMM dirichlet multinomial mixture, PERMANOVA permutational multivariate analysis of variance, Strep *Streptococcus,* Staph *Staphylococcus.* Source data are provided as a Source data file.

this was based on the microbiomes of the anterior nares and from an adult population (Human Microbiome Project)[14], which are different from the groups and collection sample sites used in our study. In a recent study focusing on infants, ARGs were detected in 64% of nasopharynx samples[15]. However, samples in this study were enriched for streptococci, thus precluding general conclusions on the extent of ARG presence in the natural microbiome. In our study, ARGs were present in 95% of the preterm infant samples from the nasopharynx, regardless of whether they were exposed to antibiotics or not. Although our study differs in several aspects from the ones above, the combined findings indicate that the respiratory tract is likely an important reservoir of ARGs and an integral component of human respiratory microbiomes. For preterm infants, the detection of ARGs in the nasopharynx during the first few hours of life might occur due to microbiome colonization from multiple sources such as transgenerational transfer from mothers and the environment[16,17]. Procedures common in neonatal intensive care, such as intubation, feeding tube insertion, and suction catheters, may also disrupt anatomical barriers in the respiratory system, potentially altering the patterns of microbial colonization[18]. In addition, preterm infants typically necessitate extended hospital stays after birth compared to term infants, with a greater likelihood of requiring care in NICUs. This most likely contributes to an increased risk for respiratory tract colonization with drug-resistant microorganisms, as recently reported in relation to the gut resistome[19].

We found that ARGs conferring multidrug resistance via efflux pumps encompassed a large proportion of the nasopharyngeal resistome in the preterm infants. Similar findings have been previously reported in the gut and oral resistome of preterm and term infants by other metagenomic studies[7,16,20,21]. In addition, we detected many of the same aminoglycoside-modifying enzymes, tetracycline protection proteins, and beta-lactamase ARGs identified previously in the nasopharynx of South African infants, even though the samples in the South African study were culture-enriched for streptococci and were from a country with higher AMR challenges than the origin of samples used in the present study. Such similarities in resistome composition may, at least in part, be explained by the large proportion of streptococci found in the natural microbial communities of the infant nasopharynx. Together, these findings suggest the existence of a core infant nasopharyngeal resistome.

*Staphylococcus aureus* is one of the leading pathogens causing neonatal sepsis and one of the priority drug-resistant pathogens by the WHO[22]. We found a strong correlation between the abundance of *S. aureus/epidermidis* and the PC1-beta-lactamase (*blaZ*) gene.

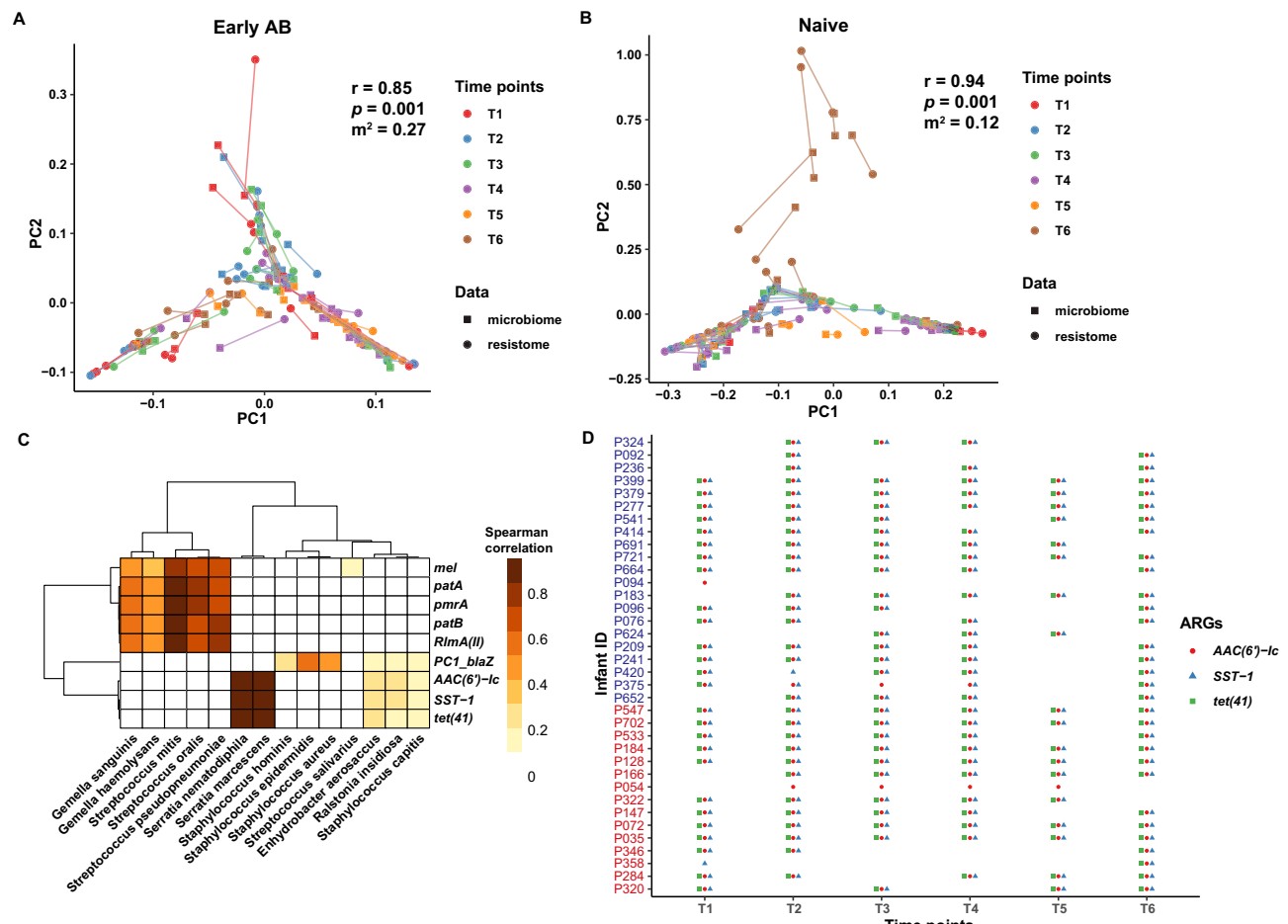

**Fig. 6 | The resistome variation in nasopharynx is explained by microbiome variation.** The ARG and microbial abundance profiles were correlated with each other for antibiotic-treated (**A**) and naive (**B**) groups using Procrustes analysis. The length of line connecting two points denotes the degree of dissimilarity or distance between the microbiome (filled square) and resistome (filled circle) composition of the same sample. The line and points are colored based on time points in both groups. The correlation coefficients (r) and significance (p) were calculated using Procrustean randomization tests (permutation-based) with the *protest* function in vegan package. The m² value represents the sum of squared distances between the matched sample pairs. **C** Heatmap depicts the Spearman's pairwise correlation coefficients (r) between individual ARGs and microbial species from nasopharyngeal samples. Hierarchical clustering using Euclidean distance was performed on both rows and columns. Correlation coefficients (r) are shown only where adjusted *P*-value (adjusted using Benjamini–Hochberg multiple test correction method) <0.05. **D** Timeline of ARGs associated with *S. marcescens* in the nasopharynx samples of each preterm infant across time points. Each ARG is represented by a different color and shape. Source data are provided as a Source data file.

This gene is prevalent in the chromosomes and mobile plasmids of Staphylococci[23], thus representing a potential high-risk ARG. Another strong correlation was found between *Serratia* species and specific resistance genes such as *SST-1*, which confers resistance against cephalosporins. Other resistance genes correlated with this group included aminoglycoside (*AAC(6)-Ic*) and tetracycline (*tet(41)*) resistance genes. Many clinical and environmental isolates of *Serratia spp.*, especially *S. marcescens* strains, have been reported to harbor these multiple ARGs on their chromosomes or plasmids[24–27]. This holds particular relevance, given that *S. marcescens* is linked to nosocomial infections and hospital outbreaks, notably in NICUs[27,28]. Multidrug-resistant *Serratia spp.* are listed among the most critical bacteria by the WHO for which innovative new treatments are urgently needed[22].

Our results showed a significant effect of early broad-spectrum antibiotic regimen (ampicillin + gentamicin) on the total abundance, diversity, and composition of ARGs in the nasopharynx of preterm infants. These effects on the resistome were short-lived and only observed directly after the cessation of antibiotic treatment (T2). In a recent study investigating the impact of antibiotics used in the first week of life on gut resistome, Reyman et al. reported that three other antibiotic combinations (penicillin + gentamicin, co-amoxiclav +

gentamicin, and amoxicillin + cefotaxime) also had a transient impact on the infant resistome[29]. As the effects of antibiotics on microbiome and resistome composition are unique to each individual and dependent on the antibiotics used and the body site, caution should be taken when extending the interpretation of such general findings. Interestingly, the transient effects of ampicillin + gentamicin on the nasopharyngeal resistome were more discernible in preterm infants whose mothers were exposed to antibiotics during pregnancy before delivery (prenatal antibiotics). Moreover, we also observed a more profound effect of prenatal antibiotic exposure on the nasopharyngeal resistome of preterm infants at T2 (after cessation of early antibiotics), which persisted until the last sampling time point before discharge from the NICU (T4-T5), but not at 6 months corrected age. Although very few studies have focused on the association between prenatal antibiotic exposure and the resistome, a recent study based on the gut indicates a possible correlation of prenatal antibiotics with increased abundance of ARGs, at least when used 5 to 6 weeks before birth[7]. In addition, in concordance with other studies on the infant gut[16,30], our data showed no significant impact of intrapartum antibiotic therapy on the overall nasopharyngeal resistome composition. The nasopharyngeal samples of preterm infants at 6 months corrected age were still

clustered according to the birth weight group (similar to T1, Supplementary Fig. 6C), suggesting a strong but transient influence of other environmental factors, such as antibiotic exposures. Nonetheless, there is a paucity of relevant studies on nasopharynx that can corroborate our findings, as they are mainly restricted to the infant gut resistome[16,17,19,29–31].

One of the strengths of this study is that it presents the most comprehensive information to date on the characteristics and dynamics of resistome development in the nasopharynx, including results for the first 8–10 months of life (6 months corrected age)–a critical period of airway development in infants. In addition, the naive group was enrolled at the same time as our treated group from the same hospital, thus removing potential confounding differences in enrollment years or sampling duration. However, the small number of samples at each time point and the inter-individual response to antibiotic exposure may be insufficient to unravel significant associations between antibiotic regimen and resistome development. Moreover, the baseline samples for the antibiotic-treated group ideally would represent samples taken before treatment. However, the initiation of antibiotic treatment for sEONS and the clinical stabilization of the preterm infants were a high priority that needed to precede nasopharynx sample collection. To partially address this issue, we included only preterm infants with samples collected no later than 24 h after antibiotic initiation. Even so, this choice may have resulted in the selection of more clinically stable infants, which could potentially limit the generalizability of our results. Despite this approach, the differences in resistome changes between baseline and after treatment found in our study may underestimate the true impact of antibiotics on the resistome. We rigorously adjusted for such confounding factors and inter-individual variation using multivariable methods or matched analysis to ensure the reliability of our observations as a consequence of early antibiotics. Nevertheless, we cannot isolate early antibiotic effects from other adverse pre-life events that coincide with premature birth, such as prenatal and antepartum maternally administered antibiotics. Furthermore, for low biomass samples such as those used in our study, sensitive approaches such as deep sequencing are necessary to obtain meaningful microbiome information. This increases the risk of detecting potential DNA contaminants. We mitigated these challenges by implementing rigorous measures spanning from sample collection to data analysis. This included the concurrent collection of samples from treated and naive groups, stringent protection measures during sample collection, incorporation of negative and positive controls, and meticulous in silico identification and removal of possible contaminants using DNA sequencing data.

In conclusion, our findings suggest that early-life treatment with broad-spectrum antibiotics for sEONS can cause perturbations in the nasopharyngeal resistome of preterm infants. However, the overall perturbations were transient, as we did not observe a significant alteration in the diversity, abundance, composition, or carriage persistence of potentially high-risk ARGs in treated preterm infants compared to controls at 6 months of corrected age. Despite this, such findings hold significance, considering the vulnerability of preterm infants in their first days of life. The individual variability in the distribution of high-risk ARGs and the changes in the resistome following antibiotic treatment highlight the potential benefits of developing resistome profiling tools for point-of-care use, which could aid in informing and tailoring antibiotic treatment decisions[32]. In addition, prolonged carriage of ARGs associated with nosocomial S. marcescens was noted. Future multicenter studies with long-term follow-up are needed to assess the ecological side-effects of hospitalization and different antibiotic regimens, including alternatives to broad-spectrum antimicrobial therapy on infant respiratory resistome development. Understanding the development of the respiratory resistome and the impact of antibiotic therapies is relevant for striking a balance between the conflicting goals of minimizing the risk of serious infectious complications to

preterm birth on one hand, and minimizing the selection of organisms with ARGs on the other. In general, the disproportionate contribution of respiratory pathogens to drug-resistant deaths at all ages highlights the importance of future studies focusing on this yet relatively unexplored reservoir of pathogens and ARGs.

## Methods

### Ethics statement
Premature infant metadata and all nasopharynx aspirate samples used in this study were collected at the Neonatal Intensive Care Unit (NICU) at Oslo University Hospital, Ullevål, Norway, as part of the Born in the Twilight of Antibiotic project. The study received approval from the Regional Committee for Medical and Health Research Ethics – South East, Norway (2018/1381 REKD), and followed the principles of the Declaration of Helsinki guidelines. Infants were enrolled in the study after obtaining written informed consent from their parents.

### Study design and cohort characteristics
The study is a prospective observational cohort study. Between July 2019 and January 2021, we included preterm infants born (or transferred within 48 h) at the NICU (Oslo University Hospital, Ullevål) who were between 28 + 0 and 31 + 6 weeks of gestational age (GA). In total, 66 preterm infants were enrolled in the study, and followed from birth until six months corrected age (i.e., adjusted for prematurity) during their stay in the NICU and after discharge. Five of the infants were lost to follow-up. Extensive metadata and covariates from infants and mothers, including mode of delivery, gestational age (GA) at birth, postmenstrual age (PMA), sex, birth weight (BW), postnatal infant antibiotics usage (duration, type, days), maternal antibiotic usage, feeding regimen during hospitalization and after discharge, and other demographic characteristics, were collected through medical records. This data was stored in a secured server at the Services for Sensitive Data (TSD), University of Oslo.

In our cohort of 66 preterm infants, 33 infants with sEONS received broad-spectrum antibiotics, consisting of a combination of ampicillin (a beta-lactam antibiotic) and gentamicin (an aminoglycoside antibiotic), within 24 h after birth. Due to the circumstances of premature birth and the instability of the infants, baseline samples could not be collected before antibiotic therapy initiation. Instead, the first nasopharyngeal aspirate samples (baseline) were obtained within 48 h after the initiation of early antibiotic therapy. To investigate temporal changes in the resistome in response to antibiotics and control for potential baseline differences as confounding factors, this paper included only infants who had nasopharyngeal samples collected on the same day as the initiation of early antibiotic treatment ($n = 15/33$) and antibiotic-naive preterm infants as controls ($n = 21$). The average early antibiotic treatment duration was 5 days (SD: 2 days). The sampling time point T2 corresponds to the first available sample collected (except in one infant) after the cessation of early antibiotic treatment. In addition to early antibiotics, one of the included infants received subsequent antibiotics initiated >72 h after birth and before discharge from NICU and two other infants received antibiotic therapy between discharge from NICU and 6 months corrected age. A timeline of sample collection and antibiotic administration is represented in Supplementary Fig. 1.

Infants exposed to early antibiotics had significantly lower GA and BW, and their mothers had a more prolonged period of ruptured membranes before delivery compared to antibiotic-naive infants. None of the mothers of naive infants received antibiotics during pregnancy (prenatal antibiotics) before onset of delivery. The baseline characteristics of the infants stratified by antibiotic exposure are presented in Table 1.

### Sample collection
Nasopharyngeal aspirate samples were collected from infants at six distinct time points spanning from birth to 6 months corrected age

**Table 1 | Cohort characteristics of preterm infants analyzed in this study**

| NP cohort | Naive | Early-antibiotics | p |
|---|---|---|---|
| n | 21 | 15 | |
| Male, n (%) | 12 (57%) | 11 (73%) | 0.484 |
| GA, weeks (mean, SD) | 31, 6/7 | 29 5/7, 1 | **<0.001** |
| BW, grams (mean, SD) | 1562, 188 | 1306, 277 | **<0.05 (0.0022)** |
| **BW, group, n (%)** | | | **<0.05 (0.013)** |
| ELBW | 0 (0%) | 2 (13%) | |
| VLBW | 8 (38%) | 10 (67%) | |
| LBW | 13 (62%) | 3 (20%) | |
| **Mode of delivery, n (%)** | | | 0.694 |
| Vaginal | 4 (19%) | 4 (27%) | |
| C-section | 17 (81%) | 11 (73%) | |
| **Antibiotics during pregnancy, n (%)** | | | **<0.001** |
| None | 21 (100%) | 6 (40%) | |
| <10 days | 0 | 7 (47%) | |
| >= 10 days | 0 | 2 (13%) | |
| **Intrapartum antibiotics (IAP), n (%)** | | | 0.50 |
| None documented | 2 (10%) | 0 (0%) | |
| Given | 19 (90%) | 15 (100%) | |
| ROM, hours (mean, SD) | 3, 12 | 432, 733 | **<0.05 (0.0108)** |
| Apgar10 (mean, SD) | 9, 1 | 8, 1 | 0.3039 |
| **Nutrition (discharge to 6 months CA)** | | | **<0.05 (0.039)** |
| Fully breastfed | 7 (33%) | 4 (29%) | |
| At least 50% mother milk, but also some formula | 3 (14%) | 6 (42%) | |
| Mostly formula and some mothers milk | 6 (29%) | 0 (0%) | |
| Formula | 2 (10%) | 4 (29%) | |
| Lost to follow-up | 3 (14%) | 1 (7%) | 0.626 |

Baseline characteristics are represented as counts (n) and percentages (%) for categorical variables, and mean and standard deviation (SD) for continuous variables. Pearson's chi-squared test or Fisher's exact test was used for categorical variables, and a two-sided unpaired t-test was used for continuous variables to calculate differences between groups (p < 0.05). P-values (p) written in bold indicate nominally significant differences between groups.

(T1–T6). The days of life (DOL) at which samples were collected for each time point varied as follows: T1 (DOL: 0–3), T2 (DOL: 4–9), T3 (DOL: 12–19), T4 (DOL: 20–31), T5 (DOL: 32–76), and T6 at 6 months corrected age (DOL: 229–288). Additional samples were taken if the antibiotic therapy was initiated or terminated more than 48 h before/ after a predefined sampling time. As a result, in two infants, there was more than one sample available for the defined time points. To facilitate time point-wise analysis, only one sample was included for each of the time points (Supplementary Data 1 and Supplementary Fig. 1). In cases where the sample could not be collected at the scheduled time points (such as with an unstable infant), a new sample was obtained as soon as possible. Infants who were transferred to another hospital before discharged to home adhered to the study protocol. When the infants reached six months of corrected age, they were either invited to an outpatient visit at Ullevål hospital or their local hospital, or they were visited at home for sample collection. Nasopharynx aspirates were collected as previously described[8], then stored and cryopreserved in 20% glycerol (2 ml) at −80 °C until further processing. In total, we collected and extracted metagenomic DNA from 198 nasopharyngeal samples obtained from 36 preterm infants, from the day of birth until 6 months corrected age. Detailed information on sample exclusion and inclusion statistics at different steps from sampling to whole metagenomic sequencing (WMS) is described in Supplementary Table 1.

**Metagenomic DNA extraction, quantification, and sequencing**

Metagenomic DNA was extracted by first thawing the aspirate samples (2 ml) on ice, pelleting them by centrifugation at $10,000 \times g$ for 10 min at 4 °C, and then using MolYsis™ for host DNA depletion (Molzym, Bremen, Germany), following the manufacturer's protocol with some modifications, as previously described[8]. After depletion of human DNA, fresh pellets were spiked with 20 µl of ZymoBIOMICS Spike-in Control II (catalog number: D6321 & D6321-10), and bacterial DNA was extracted using the MasterPure™ Gram Positive DNA Purification Kit (Epicentre, Madison, WI, USA), according to the manufacturer's recommendations. This kit employs a chemical lysis method for hard-to-lyse bacteria that has been shown to outperform mechanical lysis methods for reliable recovery of DNA from low biomass preterm nasopharyngeal samples[8]. One glycerol sample from each batch used for the cryopreservation of nasopharyngeal samples (n = 5; sampling blanks) and one reagent blank per extraction batch (n = 5; DNA extraction blanks) were extracted alongside the cohort samples using the same protocol, serving as negative controls. As positive controls, six aliquots (20 µL each) of Spike-in Control II Low Microbial Load (ZymoBIOMICS™, Catalog Nos. D6321 & D6321-10), extracted in the same manner, were included. The ZymoBIOMICS Spike-in Control II comprises of three bacterial species not found in the human microbiome (*Truepera radiovictrix*, *Imtechella halotolerans*, and *Allobacillus halotolerans*). Finally, the DNA pellet was eluted in 35 µl of TE buffer and stored at −80 °C until further use.

The total DNA concentration was quantified using Qubit dsDNA HS assay kits in a Qubit 4.0 Fluorometer (Invitrogen, Thermo Fisher Scientific, USA). Real-time quantitative polymerase chain reaction (RT-qPCR) assays were used to quantify human (Zymo E2005) and microbial DNA (Zymo E2006) in all extracted samples using Zymo Femto™ Quantification kits (ZymoBIOMICS™). The RT-qPCR results for bacterial DNA (16S rRNA) were also used to exclude samples with low or negligible bacterial load before sequencing. This was accomplished by subtracting the average yield of bacterial DNA from the positive spike-in controls from the yields of patient samples. Samples that exhibited a negative yield in RT-qPCR (n = 4) were excluded from further processing (Supplementary Data 3). The DNA yield from negative controls was extremely low or undetected (n = 10; mean = 0.00043 ng/µL), as measured by RT-qPCR. These samples were included in sequencing runs for in silico identification and removal of contaminant signals, alongside the spike-in positive controls.

Metagenomic sequencing libraries were constructed using the Nextera DNA Flex library preparation kit (Illumina Inc., CA, USA). All samples were subjected to paired-end sequencing at $2 \times 150$ bp on an Illumina NovaSeq S4 high-output platform at the Norwegian Sequencing Centre (Oslo, Norway). In total, 6.14 billion raw sequencing reads were obtained from the nasopharyngeal samples of preterm infants, with a mean sample read counts of 33.97 million (M) ranging from 4.19 to 69.25 M reads. Conversely, negative control samples exhibited an average read count of 0.15 million (M), with a range of 24,113 to 355,861 reads (Supplementary Data 1).

**Controlling for contaminant in low microbial biomass samples**

To mitigate biases due to contamination—a critical consideration in low microbial biomass microbiome research[33]—we implemented several recommendations as follows: (1) samples from infants in both groups were collected concurrently to ensure temporal consistency,

(2) standard protection measures were strictly adhered to during sample collection, including the use of clean suits, disposable gloves, and face masks, (3) both sampling and DNA extraction blanks were included as negative controls, (4) the DNA extraction and library preparation processes were conducted in a meticulously cleaned environment, with personnel wearing laboratory coats and disposable gloves, (5) longitudinal samples from each preterm infant were processed in a single extraction run, (6) spike-in using non-human bacterial species were used as positive controls. The in silico identification and removal of possible contaminants using DNA sequencing data is described below.

### Bioinformatics processing

The raw metagenomic sequencing data was pre-processed using our in-house bioinformatics pipelines using a high-performance computing cluster inside a secured environment, i.e., TSD at the University of Oslo. In brief, the Nextera adaptor sequences and low-quality reads were filtered using Trim Galore (v.0.6.1) with default parameters[34]. Next, the human DNA contaminant sequences were identified and removed by mapping the quality-trimmed reads against the human reference genome (GRCh38) using Bowtie2 (v.2.3.4.2)[35] (non-default parameters: *q -N 1 -k 1 --fr --end-to-end --phred33 --very-sensitive --no-discordant*) along with SAMtools (v.1.9)[36] and BEDTools (v.2.27.1)[37]. On average, 46.71% (range: 1.09 – 97.69%) of total quality-filtered reads were identified as belonging to the human genome. After removal of the human-associated metagenomic reads, a total of 2.98 billion remaining high-quality clean reads across all nasopharyngeal samples with number of reads ranging from 0.76 to 59.9 M per sample, and with median of 16.4 M reads were subjected to resistome profiling. More detailed results regarding the overall sequence preprocessing are presented in Supplementary Data 1. The sequence quality reports on raw and processed reads were generated using FastQC (v.0.11.9)[38] and summarized using MultiQC (v.1.7)[39].

### Resistome and microbiome profiling

To characterize the presence of ARGs, cleaned high-quality reads for each sample were mapped against the *nucleotide_fasta_protein_homolog_model* from the Comprehensive Antibiotic Resistance Database (CARD) (v.3.2.2)[23] using Bowtie2 under parameter *--very-sensitive-local*. The mapped reads from each sample were then filtered, sorted, and indexed using SAMtools. The number of reads mapped to each ARG was calculated using SAMtools *idxstats* and BEDTools. We considered ARGs with a gene fraction of at least 80% (meaning, 80% or more of the nucleotides in the ARG sequence were covered by at least one read) as positively detected in a sample. In addition, we excluded ARGs from subsequent downstream analysis if they could not be confidently attributed to resistance based solely on a short-read marker, such as those linked to resistance through mutations, as detailed by D'Souza et al.[40]. The mapped read counts were normalized for bacterial sequence abundances and gene lengths by calculating reads per kilobase of reference gene per million bacterial reads (RPKM) for each CARD reference sequence. The relative abundance of ARGs for each sample was computed by dividing the RPKM by the total sum of the RPKM for each sample. For keeping the acyclic hierarchical annotation structure for accurate resistome profiling at higher functional levels, ARGs were manually re-annotated based on the drug class to which they confer resistance. ARGs belonging to macrolides, lincosamides, and streptogramins were congregated into the MLS class. While ARGs belonging to carbapenem, cephamycin, cephalosporin, penem, penam, and monobactam were congregated into the Beta-lactam class.

Rarefaction analysis was performed using Rarefaction Analyzer[41] to assess whether the samples had sufficient sequencing depth for the comprehensive characterization of the nasopharyngeal resistome composition in preterm infants (Supplementary Fig. 3). This analysis involved the random subsampling of sequencing reads at proportions varying from 0 to 100% of the total sequencing depth. The subsampled metagenomes were not utilized for any further downstream analysis. Ten samples without identified ARGs were excluded from downstream analysis (Supplementary Table 1). These samples were collected from 9 different infants at various time points and exhibited varying numbers of clean metagenomic reads, as detailed in Supplementary Data 1. Collectively, we assessed the resistome of the nasopharyngeal microbiome in a total of 181 samples collected from 36 preterm infants.

To profile the nasopharyngeal microbiota composition of preterm infants, we utilized MetaPhlAn 3.0 software[42] (Supplementary Data 4). Since *Allobacillus halotolerans*, a species included in the ZymoBIOMICS spike-in, was not initially represented in the default MetaPhlAn database (*mpa_v30_CHOCOPhlAn_201901*), we took an additional step to incorporate its clade-specific marker genes into the database. Next, for the in silico identification of possible contaminants, we employed both the prevalence and frequency-based (combined) methods of the decontam (v.1.16.0) R package[43] with the default threshold (p-score = 0.1) on the raw microbiome and resistome count abundance tables. Contaminants identification was performed independently within each batch of glycerol and DNA extraction kit used for processing the nasopharyngeal samples from preterm infants. In total, 28 microbial species, including the three ZymoBIOMICS spike-in species, and 4 ARGs identified as contaminants, were filtered out from abundance tables prior to any downstream analysis (Supplementary Data 5).

### Statistical analysis and data visualization

All statistical analyses were conducted in R (v.4.2.1) within RStudio (v.2022.07.2 + 576)[44,45]. The ARG abundance, annotation table, and metadata file were compiled into a single data object using the phyloseq (v.1.40.0) package[46]. Figures, unless stated otherwise, were created using the ggplot2 (v.3.4.0) package[47] and further edited using Adobe Illustrator (v.16.0.0). α- and β-diversity analysis was performed using the vegan (v.2.6.2)[48] and phyloseq R packages. α-diversity was calculated using the Shannon diversity metric, and differences over time points in groups were assessed using a linear mixed-effects (LME) model with infants set as a random effect while correcting for age. The overall resistome composition of nasopharyngeal samples was visualized using principal component analysis (PCA) ordination plots. The ordinates were based on the Aitchison distance with centered log-ratio (CLR) transformed count abundance data. The variance explained by clinical covariates in the overall resistome composition by clinical covariates was statistically evaluated primarily with the Permutational multivariate analysis of variance (PERMANOVA) test using the *adonis2* function (vegan package) with 999 permutations. PERMANOVA is a powerful test used to analyze the effects of both categorical and/or continuously distributed explanatory variables. It can accommodate complex study designs, including multiple confounding factors as present in our study. However, it is not particularly sensitive to heterogeneity in dispersion between the groups. To test the homogeneity of multivariate dispersion, the permutational analysis of multivariate dispersion (PERMDISP) test was used. In case of heterogeneity, the analysis of similarities (ANOSIM) test using the *anosim* function (vegan package) was also employed to statistically re-assess the univariate association of variables with the overall resistome composition. In the multivariable, temporal PERMANOVA analyses, only the variables that individually showed a significant association with resistome composition were included.

To identify the resistotypes, the Dirichlet multinomial mixture (DMM) model approach was utilized using the DirichletMultinomial (v.1.38.0) R package[49]. Procrustes analysis was performed to determine the association between microbiota and resistome composition based on PCA using CLR-transformed ARG and species abundance profiles. The symmetric Procrustes correlation coefficients between the ordinations and *P*-values (*p*) were obtained through the *protest* function

from the vegan package. Spearman's correlation was applied to resistome and microbiome relative abundance profiles for paired samples. To avoid the potential bias introduced by Spearman's rank when ranking zero values, we removed ARGs and species that were present in less than half of the samples. The correlation coefficient and corresponding adjusted P-values (*p.adj*) were calculated using the *associate* function from the microbiome (v.1.18.0) package[50]. Associations between individual taxa and overall resistome outcomes were calculated using repeated measure correlation analysis from the rmcorr (v.0.5.4) package while controlling for inter-individual variation[51]. Heatmaps were created using the *pheatmap* function from the pheatmap R package (v.1.0.12)[52]. For simple, independent comparisons of group differences, we used a one-way analysis of variance test, Wilcoxon rank-sum test, Kruskal–Wallis test, or chi-square test as appropriate, and we considered $p < 0.05$ to be significant. For all analyses involving multiple comparisons, we applied the Benjamini–Hochberg (BH) method to correct for multiple testing. The statistical analysis scheme used in the study is shown in Supplementary Fig. 2.

### Reporting summary
Further information on research design is available in the Nature Portfolio Reporting Summary linked to this article.

## Data availability
The sequencing data (after filtering of human DNA) for all the samples that support the findings of this studies have been made available at NCBI SRA under the accession number PRJNA1009231. Source data underlying the main as well as Supplementary Figs. are provided in this paper. Source data are provided with this paper.

## Code availability
The software packages used in this manuscript are free and open source. No custom code was used in generating this manuscript. The analysis scripts used here are available from the authors upon reasonable request.

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

## Acknowledgements

We would like to thank the parents of participating infants, and the clinical staff at the Neonatal Intensive Care units at Ullevål Hospital for their assistance with sample collection. The sequencing service for this work was provided by the Norwegian Sequencing Centre (www.sequencing.uio.no) and the computations were performed on resources provided by Sigma2 - the National Infrastructure for High Performance Computing and Data Storage in Norway. This work was financed and supported by the Research Council of Norway (RCN), project numbers: 273833 (F.C.P./A.D./P.R./G.S./H.A.A./D.B./O.D.S./U.D./K.H.), 322375 (RESISFORCE network: F.C.P./A.D./G.S./H.A.A.), 274867 (RESISPART network: F.C.P./A.D./G.S./H.A.A.), 299159 (NORSE network: U.D./F.C.P.), and the Olav Thon Foundation (F.C.P./A.D./P.R./G.S./H.A.A./D.B./O.D.S./G.G./U.D./K.H.). The Faculty of Dentistry at the University of Oslo and Oslo University Hospital also provided support.

## Author contributions

F.C.P. conceptualized and designed the study, acquired funding, and contributed to data interpretation. A.D. conducted bioinformatics and statistical analysis, performed data interpretation, and wrote the manuscript with input from F.C.P. P.R. and K.H. were responsible for patient inclusion, sampling, and metadata analysis. P.R., G.S., and H.A.A. conducted sample processing, metagenomic DNA extraction, and library preparation for sequencing. D.B., O.D.S., D.F., G.G., U.D., and K.H. contributed to study design, provided critical revisions on the manuscript. All authors contributed to manuscript review and provided valuable feedback during the revision process.

## Competing interests

The authors declare no competing interests.
