## [Peer Review File · Nature Communications]

REVIEWER COMMENTS

Reviewer #1 (Remarks to the Author):

With great interest we have read the work of Dhariwal et al. entitled "The landscape of antibiotic resistance genes in the nasopharynx of preterm infants: Prolonged signature of hospitalization and effects by antibiotics". Dhariwal and co-authors investigated a critical aspect of antibiotic resistance within the nasopharyngeal microbiome of preterm infants. Their research utilized shotgun metagenomics to comprehensively explore various aspects of the nasopharyngeal resistome, including its development, and its response to early antibiotic exposure over time. The study provides in-depth insights into the characteristics and dynamics of the nasopharyngeal resistome, over the first six months of life in preterm infants. While highly interesting and potentially of great value to its field, I am afraid it is essential to first address several pivotal concerns before we can truly understand and interpret the presented data.

- First, the authors use a novel optimized laboratory method to extract (DNA MasterPure™_Gram Positive DNA Purification Kit) and eliminate host DNA (MoLysis™). They refer to their own published method & validation paper, though from this paper it remains still unclear how previously reported problems, such as proper representation of all microbes present, is overcome, usually involving mechanical lysis besides chemical lysis.

<https://www.frontiersin.org/articles/10.3389/fmicb.2022.1038120/full>

- Also, host DNA depletion relies on type of samples and whether fresh or frozen. In the latter case, samples frozen without any stabilizing agent makes it impossible to separate host DNA out, while preserving microbial DNA. There is no detailed information on sample storage in the manuscript nor in the methods paper.

- It is unclear which methods were employed for microbial taxonomy assignment and the specific databases used for this purpose. While acknowledging that this information may be available in a upcoming article, it is still crucial to provide a brief overview of how this was done.

- Additionally and importantly, it is currently unclear whether any pre-processing (e.g., removal of low-quality samples, identification of possible contaminants) steps were utilized in regards to the microbial community. For instance, it would be valuable to know why negative/positive controls were not utilized in downstream analyses to address contamination issues. This is particularly important in low biomass nasopharyngeal samples, as contamination is always impacting the data, despite laboratory cleanliness and the use of negative controls (intrinsic to the type of samples and procedures).

- In addition, authors did not report sample information on bacterial biomass, which would have been helpful to assess the risk of contamination in these samples. Also, quantification after extraction and host DNA removal was done on total DNA rather than microbial DNA, which was used for pooling of samples for sequencing. As the table shows this has resulted in very variable microbial reads, sometimes

really low, certainly when focusing in the analyses on ARGs only. There is in our opinion a risk of skewing of data due to 'non-equimolar' pooling on microbial level. Quantification of microbial DNA to ensure sufficient input DNA and exclude too low biomass samples, would have been advisable.

- As data still suggest very low microbial biomass, a decontamination step in identifying the true biological signal, and eliminating the contaminating reads, just can't be omitted. Stating 'the

- Authors explain that they followed recent recommendations for low microbial biomass microbiome research, e.g. (1) sampling groups within the same time period, (2) using standard protection measures during sample collection, and (4) laboratory procedures, and (5) using unique barcodes in library preparation. These practices, regardless of the sample's biomass, are common and standard in microbiome research. Though indeed essential for the type of research, these on their own are not sufficient to address the issue of contamination. Stating "the negative controls had no or extremely low DNA concentrations to proceed to library preparation as determined by RT-PCR were therefore excluded from further analysis" is not sufficient. Proactive attempts should be made to identify contaminating signals (especially the signals that will be created by laboratory processing), which actually can be done by attempting to sequence blanks with limited signal (equimolar pooling) AND using positive controls; not spike-in controls, but Zymo mocks themselves, as it is known what should be in there, so the 'additional signal that entered the controls' can be used to identify contamination (see for example Salter et al. BMC biology 2014).

- sufficient number of reagent controls and/or density measurements allows for the use of R-packages like Decontam (Davis, Microbiome, 2018), ensuring detailed filtering of (potential) contaminating taxa, which are crucial for studies involving low biomass samples as contaminating signal may overpower the true biological one.

- The approaches described in the methodology section (such as their motivation for using non-parametric vs parametric tests) do not always align with the findings presented in the results section/figures. In the section "Statistical analysis and data visualization", the authors describe using PERMANOVA tests to assess the association between the resistome and various covariates. They also included PERMDISP analyses to assess the homogeneity of multivariable dispersion, which they included to verify the assumptions of the PERMANOVA test. However, there are no figures or tables (neither in the supplement) that includes the results of the PERMDISP analyses. The study would benefit from further clarification and transparency for their choice of using PERMANOVA over ANOSIM, and the accompanying data).

- In the section "Resistome profiling", the authors describe the approach used for characterizing ARGs by utilizing the CARD database. However, to fully appreciate these results, one should study and interpret these (also the reader) in the context of microbial community composition data, which are lacking from this picture. The authors explain those data are currently part of a different manuscript which is under review. This makes it impossible to interpret the data, its value, accuracy, and impact. We would strongly advise adding these data to the manuscript.

- The choices for certain statistical methods (such as the use of different ordination methods) used are not always well explained. For instance, the section "Statistical analysis and data visualization" two different ordination methods are described. Specifically, when assessing the overall resistome composition and its relationship with various covariates (host factors), the authors utilized PCA

ordination plots based on a Euclidean dissimilarity matrix of CLR transformed relative abundances data. Conversely, when comparing the resistome composition with the microbiome composition, they employed PCoA ordination plots based on a Bray-Curtis dissimilarity index. The authors should provide an explanation in this section regarding the rationale behind using these two different ordination methods.

- The manuscript would greatly benefit from a thorough proofreading, as it includes numerous grammatical errors and incomplete or unclear sentences that need attention. Furthermore, the notation of results, such as p-values and R² values, lacks consistency throughout the manuscript.

Reviewer #2 (Remarks to the Author):

Summary

In their manuscript, Dhariwal et al. note that the upper respiratory tract, specifically the nasopharynx, is often colonized by potential pathogens. The authors set out to characterize the resistome of this community in preterm infants using shotgun sequencing. The cohort was chosen in part due to their unique vulnerability to infection and also to map out the impact of antibiotic treatment, which is common in preterm infants, on the nasopharyngeal resistome. Assembly and management of this cohort and its samples is an impressive accomplishment. The authors also looked at clinical outcomes in this context. The authors deserve credit for observing current best practices in working with low biomass samples. The manuscript is well written, and its findings are justified by the presented results.

Major findings

The authors observed that essentially all nasopharyngeal samples encoded ARGs and found a significant association between beta lactam and aminoglycoside treatment and increased total ARG abundance. In the treated cohort compared to untreated, the authors show that there is an increase in beta lactam and aminoglycoside resistance-specific mechanisms and ARGs as well, though in total the most common mechanism appears to be less specialized efflux pumps. Treatment also resulted in increased carriage of ARGs until around two to three weeks post-treatment, at which point ARG abundance became indistinguishable from pre-treatment levels. This pattern of significant but transient perturbation was found across a variety of analyses. In treated vs untreated infants, the authors found that ARG composition (not abundance) correlated with birth weight rather than treatment (except immediately following treatment). The authors did not find as significant results when looking at covariation with other cohort variables. The authors also used unsupervised learning to classify resistomes into resistotypes, finding that the 3 resulting types were distinguished by fluoroquinolone resistance and ribosome methylation (RlmA(II)), beta lactamase and tetracycline efflux (Tet(41)), and another beta lactamase (BlaZ). Community analysis found that these resistotypes were strongly correlated with the

presence of *Streptococcus* (likely RImA host), *Serratia* (likely Tet(41) host), and *Staphylococcus* (likely BlaZ host). The authors found that carriage of *Serratia*-linked ARGs persisted across many timepoints and point to a concurrent *Serratia* outbreak at the hospital, but they do not follow this thread further. The authors note that nasopharyngeal microbiome studies are less common than gut microbiome studies and have made an important contribution to this field.

Major comments

- Reviewer does not note any significant issues

Minor comments

- (lines 103 to 122) The details of the cohort could benefit from a bit more details. First, I think it is implied but I did not see explicitly stated that the no-antibiotic control were also preterm births? And it appears that for treated infants, only those with sampling within 24 hrs of treatment onset were analyzed (I believe), but the methods section does not make it clear that this meant only 15 infants out of the 33 with sEONS antibiotic treatment were followed (unless I misunderstood).
- (line 128) The authors mention that some timepoints had more than one sample due to additional antibiotic treatment within 48 hrs, did the authors have a system for choosing which timepoint to use in those cases or was it randomly decided?
- (line 211) For the 10 samples with no ARG reads assigned, did they come from the same patient, timepoints etc? Did they have lower sequencing depth? It seems potentially interesting that there were 0 ARGs associated with them, unless a more mundane technical explanation is available.
- The authors mention a few times that previous studies of early life resistomes have focused on the gut microbiome but they do not compare their nasopharyngeal findings to these examples. I don't think that a close comparison of the data are needed, but how do broad trends differ between these environments (e.g., they report transient resistome changes here, but many prior gut microbiome studies find longer term effects, does this reflect variance in patients or is the nasopharyngeal microbiome less/more robust to change?)
- (line 424) I feel like this line is a bit of a tease! Is the outbreak *Serratia* strain sequenced/available? Can you map shotgun ARG and non-ARG reads to it?
- (line 474) The authors mention transgenerational transfer of microbiomes from mother to infant and the environment as sources of microbiomes. Don't many infants in NICU have feeding tubes and intranasal cannula? It seems like these invasive surfaces could also be important and cohort-specific sources of hospital-associated microbes.
- In a few sections the authors try to put the detected ARGs in a risk framework. I think this would be more compelling if there was data on mobile genetic elements to back it up. Can the authors detect syntenic MGEs next to ARGs? Looking at the methods, it appears that the shotgun reads were not actually assembled, do you think it would be worth doing?

Reviewer #3 (Remarks to the Author):

REVIEWER COMMENTS

Reviewer #1 (Remarks to the Author):

With great interest we have read the work of Dhariwal et al. entitled "The landscape of antibiotic resistance genes in the nasopharynx of preterm infants: Prolonged signature of hospitalization and effects by antibiotics". Dhariwal and co-authors investigated a critical aspect of antibiotic resistance within the nasopharyngeal microbiome of preterm infants. Their research utilized shotgun metagenomics to comprehensively explore various aspects of the nasopharyngeal resistome, including its development, and its response to early antibiotic exposure over time. The study provides in-depth insights into the characteristics and dynamics of the nasopharyngeal resistome, over the first six months of life in preterm infants. While highly interesting and potentially of great value to its field, I am afraid it is essential to first address several pivotal concerns before we can truly understand and interpret the presented data.

Thank you for your valuable feedback. We appreciate your interest in our work. and fully acknowledge the importance of addressing the specified concerns.

- First, the authors use a novel optimized laboratory method to extract (DNA MasterPure™ _Gram Positive DNA Purification Kit) and eliminate host DNA (MoLYsis™). They refer to their own published method & validation paper, though from this paper it remains still unclear how previously reported problems, such as proper representation of all microbes present, is overcome, usually involving mechanical lysis besides chemical lysis. <https://www.frontiersin.org/articles/10.3389/fmicb.2022.1038120/full>

Response: Thank you for your comment regarding the potential benefits of mechanical lysis. This is generally well documented for gut microbiome studies. However, our study focused on preterm nasopharyngeal samples, which present unique challenges due to their low microbial biomass. This holds true not only for 16S microbiome studies but even more so for whole metagenomic sequencing, as the latter often necessitates larger amounts of DNA.

Despite our attempts to use hybrid methods combining mechanical and chemical lysis, namely MagMax and QIAamp protocols, we were unable to reliably retrieve sufficient DNA from both nasopharyngeal samples and low biomass mock controls. As a result, we decided to use the chemical lysis method that addresses difficult to lyse bacteria, which we validated and found to reliably recover DNA from preterm nasopharyngeal samples (<https://www.frontiersin.org/articles/10.3389/fmicb.2022.1038120/full>).

While we agree that there is a need for further methodological studies, we believe that our study contributes to the existing knowledge by providing unprecedented insights into the nasopharyngeal resistome. The rationale for the method used is now included in the manuscript (Lines: 485-487).

- Also, host DNA depletion relies on type of samples and whether fresh or frozen. In the latter case, samples frozen without any stabilizing agent makes it impossible to separate host DNA out, while preserving microbial DNA. There is no detailed information on sample storage in the manuscript nor in the methods paper.

Response: Thank you for bringing this up. We would like to clarify that we preserved our samples in glycerol, which is commonly used as a cryoprotectant to prevent lysis of cells during freezing and thawing and to extend sample storage time.

Specifically, in our method paper we stated that "A sterile 2 ml 20% glycerol solution was suctioned directly afterwards through the suction catheter to rinse any mucus remains and for cryopreservation of the sample. The samples were rapidly moved to -80°C , where they were stored for up to 10 months." This is now also mentioned in the current manuscript (Lines: 470-471).

- It is unclear which methods were employed for microbial taxonomy assignment and the specific databases used for this purpose. While acknowledging that this information may be available in an upcoming article, it is still crucial to provide a brief overview of how this was done.

Response: Thank you for this comment. We agree and have now added such an overview in the method section (Lines: 580-584).

- Additionally, and importantly, it is currently unclear whether any pre-processing (e.g., removal of low-quality samples, identification of possible contaminants) steps were utilized in regards to the microbial community. For instance, it would be valuable to know why negative/positive controls were not utilized in downstream analyses to address contamination issues. This is particularly important in low biomass nasopharyngeal samples, as contamination is always impacting the data, despite laboratory cleanliness and the use of negative controls (intrinsic to the type of samples and procedures).

Response: Thank you for pointing this out. We have now sequenced all negative and positive controls for *in-silico* identification of possible contaminants in both microbiome and resistome data. We utilized the decontam R package, utilizing both "frequency" and "prevalence" methods, to meticulously identify possible contaminants in microbiome and resistome data.

- In addition, authors did not report sample information on bacterial biomass, which would have been helpful to assess the risk of contamination in these samples. Also, quantification after extraction and host DNA removal was done on total DNA rather than microbial DNA, which was used for pooling of samples for sequencing. As the table shows this has resulted in very variable microbial reads, sometimes really low, certainly when focusing in the analyses on ARGs only. There is in our opinion a risk of skewing of data due to 'non-equimolar' pooling on microbial level. Quantification of microbial DNA to ensure sufficient input DNA and exclude too low biomass samples, would have been advisable.

Response: We appreciate the constructive feedback. We have now sequenced the negative (n=10) and positive spike-in (n=6) controls, which we believe add to the robustness of the study. The sequencing results are detailed in Supplementary Table 2. Additionally, we have incorporated RT-PCR microbial DNA quantification data for each sample into Supplementary Table 4. This data was pivotal in refining our sample set, as we excluded low biomass samples by deducting the average yield of bacterial DNA in the patient samples from the spike-in controls, as to omit samples with negative net values from subsequent analysis. We also employed RPKM for data normalization to mitigate variations in read length and sequencing depth, thereby enhancing the comparability of our results. Although we recognize that in metagenomic analysis of low microbial biomass samples, some degree of technical bias is inevitable, we believe that the rigorous controls,

combined with our comprehensive sequencing strategy, greatly enhance the reliability of our findings and provide a valuable contribution to the field.

- As data still suggest very low microbial biomass, a decontamination step in identifying the true biological signal, and eliminating the contaminating reads, just can't be omitted. Stating 'the Authors explain that they followed recent recommendations for low microbial biomass microbiome research, e.g. (1) sampling groups within the same time period, (2) using standard protection measures during sample collection, and (4) laboratory procedures, and (5) using unique barcodes in library preparation. These practices, regardless of the sample's biomass, are common and standard in microbiome research. Though indeed essential for the type of research, these on their own are not sufficient to address the issue of contamination. Stating "the negative controls had no or extremely low DNA concentrations to proceed to library preparation as determined by RT-PCR were therefore excluded from further analysis" is not sufficient. Proactive attempts should be made to identify contaminating signals (especially the signals that will be created by laboratory processing), which actually can be done by attempting to sequence blanks with limited signal (equimolar pooling) AND using positive controls; not spike-in controls, but Zymo mocks themselves, as it is known what should be in there, so the 'additional signal that entered the controls' can be used to identify contamination (see for example Salter et al. BMC biology 2014).

Response: Thank you for bringing up these important points. We have now sequenced the blanks and the positive controls (see above response). The later were Zymo Spike-in mocks, comprising *Truepera radiovictrix*, *Imtechella halotolerans*, and *Allobacillus halotolerans*, as now specified in Lines: 487-495.

- sufficient number of reagent controls and/or density measurements allows for the use of R-packages like Decontam (Davis, Microbiome, 2018), ensuring detailed filtering of (potential) contaminating taxa, which are crucial for studies involving low biomass samples as contaminating signal may overpower the true biological one.
Response: As per suggestion, we have now used the decontam R package to identify potential contaminant features (taxa and ARGs) in the microbiome and resistome data. The results of decontam are provided in the Supplementary Table 6. Also, we have revised all the downstream analysis results (figures and statistics) according to the filtered count abundance tables.
- The approaches described in the methodology section (such as their motivation for using non-parametric vs parametric tests) do not always align with the findings presented in the results section/figures. In the section "Statistical analysis and data visualization", the authors describe using PERMANOVA tests to assess the association between the resistome and various covariates. They also included PERMDISP analyses to assess the homogeneity of multivariable dispersion, which they included to verify the assumptions of the PERMANOVA test. However, there are no figures or tables (neither in the supplement) that includes the results of the PERMDISP analyses.
Response: Thank you for the comment. The results for PERMDISP to verify the assumptions of the PERMANOVA test are now presented in Supplementary Table 3, and its use clarified in the workflow Supplementary Figure 2.
- The study would benefit from further clarification and transparency for their choice of using PERMANOVA over ANOSIM, and the accompanying data).

Response: The primary reason for selecting PERMANOVA over ANOSIM is its robustness in handling multiple potential confounding variables, including both categorical and continuous variables, as well as within-sample correlation (repeated measures), which were prevalent in our study. In cases of heterogeneity, we also employed the analysis of similarities (ANOSIM) test. By using this test, we could re-assess the univariate association of variables with the overall resistome composition. For multivariate temporal analysis, PERMANOVA was always the method of choice since ANOSIM is limited in handling continuous variables and cannot effectively model confounding factors in complex study designs. We have now included the results of the ANOSIM in Supplementary Table 3, alongside the PERMANOVA and PERMADISP results, to enhance transparency. This clarification has been added to the manuscript text (Lines: 608-612).

- In the section “Resistome profiling”, the authors describe the approach used for characterizing ARGs by utilizing the CARD database. However, to fully appreciate these results, one should study and interpret these (also the reader) in the context of microbial community composition data, which are lacking from this picture. The authors explain those data are currently part of a different manuscript which is under review. This makes it impossible to interpret the data, its value, accuracy, and impact. We would strongly advise adding these data to the manuscript.

Response: Thank you for your valuable feedback. Following the agreement among various institutions and PhD students involved in the project, we decided to address the microbial community composition data in a distinct manuscript. This strategic approach was based on the premise that separating microbiome and resistome data into different publications allows for a more thorough examination of each dataset, thereby yielding clearer understanding for readers with particular research interests. This approach complements and enhances our work, similar to other recent studies concentrating on resistome within the microbiome (examples: <https://www.nature.com/articles/s41467-023-36781-w>, <https://pubmed.ncbi.nlm.nih.gov/37507809/>).

However, recognizing your concerns about providing a comprehensive analysis, we have made further amendments. We’ve enriched this manuscript by including additional microbiome information in the methods section (Lines: 580-584), supplementary figure (Supplementary Figure. 9), and abundance profile (Supplementary Table 5). We believe these extra details will present readers with a more holistic view of our research.

- The choices for certain statistical methods (such as the use of different ordination methods) used are not always well explained. For instance, the section “Statistical analysis and data visualization” two different ordination methods are described. Specifically, when assessing the overall resistome composition and its relationship with various covariates (host factors), the authors utilized PCA ordination plots based on a Euclidean dissimilarity matrix of CLR transformed relative abundances data. Conversely, when comparing the resistome composition with the microbiome composition, they employed PCoA ordination plots based on a Bray-Curtis dissimilarity index. The authors should provide an explanation in this section regarding the rationale behind using these two different ordination methods.

Response: Thank you for this comment. We have now revised the text (Lines: 621-622) and re-done the integrative analysis (figure as well as statistics) to be consistent in the ordination method (PCA) used earlier in the manuscript.

- The manuscript would greatly benefit from a thorough proofreading, as it includes numerous grammatical errors and incomplete or unclear sentences that need attention. Furthermore, the notation of results, such as p-values and R2 values, lacks consistency throughout the manuscript.

Response: We appreciate the reviewer's feedback. We have now carefully revised the manuscript for grammatical errors, incomplete sentences, and inconsistencies in the notation of results to ensure that it is clear, concise and consistent throughout the manuscript.

Reviewer #2 (Remarks to the Author):

Summary

In their manuscript, Dhariwal et al. note that the upper respiratory tract, specifically the nasopharynx, is often colonized by potential pathogens. The authors set out to characterize the resistome of this community in preterm infants using shotgun sequencing. The cohort was chosen in part due to their unique vulnerability to infection and also to map out the impact of antibiotic treatment, which is common in preterm infants, on the nasopharyngeal resistome. Assembly and management of this cohort and its samples is an impressive accomplishment. The authors also looked at clinical outcomes in this context. The authors deserve credit for observing current best practices in working with low biomass samples. The manuscript is well written, and its findings are justified by the presented results.

Major findings

The authors observed that essentially all nasopharyngeal samples encoded ARGs and found a significant association between beta lactam and aminoglycoside treatment and increased total ARG abundance. In the treated cohort compared to untreated, the authors show that there is an increase in beta lactam and aminoglycoside resistance-specific mechanisms and ARGs as well, though in total the most common mechanism appears to be less specialized efflux pumps. Treatment also resulted in increased carriage of ARGs until around two to three weeks post-treatment, at which point ARG abundance became indistinguishable from pre-treatment levels. This pattern of significant but transient perturbation was found across a variety of analyses. In treated vs untreated infants, the authors found that ARG composition (not abundance) correlated with birth weight rather than treatment (except immediately following treatment). The authors did not find as significant results when looking at covariation with other cohort variables. The authors also used unsupervised learning to classify resistomes into resistotypes, finding that the 3 resulting types were distinguished by fluoroquinolone resistance and ribosome methylation (RlmA(II)), beta lactamase and tetracycline efflux (Tet(41)), and another beta lactamase (BlaZ). Community analysis found that these resistotypes were strongly correlated with the presence of *Streptococcus* (likely RlmA host), *Serratia* (likely Tet(41) host), and *Staphylococcus* (likely BlaZ host). The authors found that carriage of *Serratia*-linked ARGs persisted across many timepoints and point to a concurrent *Serratia* outbreak at the hospital, but they do not follow this thread further. The authors note that nasopharyngeal microbiome studies are less common than gut microbiome studies and have made an important contribution to this field.

Thank you for your constructive feedback. We appreciate your interest in our work and fully acknowledge the importance of addressing the specified concerns.

Major comments

- Reviewer does not note any significant issues

Minor comments:

- (lines 103 to 122) The details of the cohort could benefit from a bit more details. First, I think it is implied but I did not see explicitly stated that the no-antibiotic control were also preterm births? And it appears that for treated infants, only those with sampling within 24 hrs of treatment onset were analyzed (I believe), but the methods section does not make it clear that this meant only 15 infants out of the 33 with sEONS antibiotic treatment were followed (unless I misunderstood).

Response: Thank you for your comment. This has now been clarified in the manuscript (Lines: 435-450).

- (line 128) The authors mention that some timepoints had more than one sample due to additional antibiotic treatment within 48 hrs, did the authors have a system for choosing which timepoint to use in those cases or was it randomly decided?

Response: No, the selection was not random. In such case, we selected the samples with higher number of clean high-quality reads after removal of human DNA as well as bacterial reads available for microbiome and resistome profiling. The sequencing read statistics on such samples are now also added in the Supplementary Table 2.

- (line 211) For the 10 samples with no ARG reads assigned, did they come from the same patient, timepoints etc? Did they have lower sequencing depth? It seems potentially interesting that there were 0 ARGs associated with them, unless a more mundane technical explanation is available.

Response: These 10 samples were from 9 different infants at different time points as mentioned in Supplementary Table 1. For some samples, such as S_54, lower sequencing depth is the primary reason for not identifying any ARGs. On the other hand, for some of these samples, despite having good sequencing depth, for example, S_239 (6.4 M clean reads), no ARGs were identified. One plausible explanation is the coverage threshold used to define the presence or absence of ARGs. When looking at the results, we found that the detected ARGs (mostly with low abundance (read counts)) were not consistently covered (<80% gene fraction) by sequencing reads in samples. In contrast, samples with lower sequencing reads may still capture some reads from these low-abundant ARGs, leading to their identification. The sequencing read statistics of these samples have now been added to Supplementary Table 2.

- The authors mention a few times that previous studies of early life resistomes have focused on the gut microbiome but they do not compare their nasopharyngeal findings to these examples. I don't think that a close comparison of the data are needed, but how do broad trends differ between these environments (e.g., they report transient resistome changes here, but many prior gut microbiome studies find longer term effects, does this reflect variance in patients or is the nasopharyngeal microbiome less/more robust to change?)

Response: We thank the reviewer for this comment. It is widely acknowledged that the impact of antibiotics on the resistome is influenced by several factors, including the state of microbiome at the time of perturbation as well as the strength of the perturbation (e.g., route, spectrum, class, duration) (Schwartz DJ et al., 2020, Thänert R et al., 2022). In addition to the differences in microbiome composition between two distinct environments (gut and nasopharynx), variations in antibiotic treatments used, subjects under treatment, and specific study methods add further complexity to compare our findings with other studies. We have

mentioned the broad trends between the two environments in the discussion section (Lines: 349-352, 361-366).

- (line 424) I feel like this line is a bit of a tease! Is the outbreak *Serratia* strain sequenced/available? Can you map shotgun ARG and non-ARG reads to it?

Response: We agree that the mentioned line may come across as teasing. Regarding the outbreak strain, it has been sequenced; however, it is not disclosed or made publicly available at this time.

- (line 474) The authors mention transgenerational transfer of microbiomes from mother to infant and the environment as sources of microbiomes. Don't many infants in NICU have feeding tubes and intranasal cannula? It seems like these invasive surfaces could also be important and cohort-specific sources of hospital-associated microbes.

Response: Most of the infants do have nasogastric feeding tubes but only a few were intubated and mostly only for a few days. We agree that these surfaces could also be an important and cohort-specific source of hospital-associated microbes. We have now mentioned it in the discussion (Lines: 315-317).

- In a few sections the authors try to put the detected ARGs in a risk framework. I think this would be more compelling if there was data on mobile genetic elements to back it up. Can the authors detect syntenic MGEs next to ARGs? Looking at the methods, it appears that the shotgun reads were not actually assembled, do you think it would be worth doing?

Response: Assembling the short sequencing reads might be a way to get the genetic context information of ARGs (ARGs located on MGE sequences). However, in addition to computationally demanding and time-consuming, a high proportion of low abundant potential ARGs and MGEs might remain undetected with assembly-based approach (Boolchandani et al., 2019). Also, MGEs often contain homopolymers, long stretches of repeated sequences which are often difficult to assemble (Lee K et al, 2021, O'Connor, L et al, 2023). So even with assembly, we would miss information on HGT. For this, we would need long read sequencing or some recently developed innovative sequencing techniques, such as Hi-C (Kent et al., 2020), epic-PCR (Spencer et al., 2016), and sequencing of CRISPR-Cas spacer sequences (Munck et al., 2020).

Since we used read-based profiling in this study, we have used the term “potential” high risk ARG when putting the detected ARGs in the risk framework in our result section (Lines: 128, 337). We have also added an explanation to the risk classification system as proposed by Zhang and colleagues, which is now commonly adopted in resistome studies (Lines: 124-128). The classification is based on the likelihood of ARGs to impact human health, starting from Rank IV for ARGs not associated with humans and thus considered the least threatening, to Rank I for ARGs that already exist in pathogens, posing the highest risk. In addition, we utilized the results of association analysis predicting the potential microbial host of ARGs as well as the information present in CARD database on how often these ARGs are located on mobile genetic elements or in the chromosome, when putting ARGs in the risk framework in our discussion section. For example, the *PC-1 blaZ* reference gene sequence was prevalent in plasmids (with perfect matches/hits) and chromosome in *S. aureus* (predicted microbial host) (Lines: 335-337).

REVIEWER COMMENTS

Reviewer #1 (Remarks to the Author):

Thank you for providing us with the opportunity to review the responses of the authors.

We are pleased to see that the authors have comprehensively addressed our concerns and suggestions in their revised manuscript, especially related to the low biomass samples, their quantification, and decontamination procedures and decision making regarding 'cleaning of' the sequence data. It has been helpful to provide easy linkage to (a summary of) the community profiles published previously, as in our opinion, the resistome cannot be interpreted without knowledge on the microbiome composition.

We firmly believe that these revisions enhance the paper's overall impact and credibility. Nevertheless, we still have a few minor points we feel may need to be addressed;

- In supplementary Figure 5, it is challenging to distinguish what represents the antibiotic group versus the naïve group. Perhaps the authors could consider labeling the group names above the figures instead of relying solely on colors.
- At line 121-122 and 169-170, the authors mention that the resistome in the antibiotic-treated group is more heterogeneous compared to the naïve group. We suggest that the authors back this claim with statistical analysis, such as comparing the beta diversity indices (pairwise distances) within the antibiotic-treated group at different time points (e.g., T1 samples compared to T2, etc.) versus within the naïve group.
- There is inconsistency in the numbers for the prenatal + early group: at line 180, this group size is 8, whereas at line 215, it is listed as 9.
- Supplementary Figure 7 could benefit from statistical analysis.
- At line 254, "We noticed that..." could be changed to "We found that..."
- In Figure 6, it is unclear what "m2" indicates.

Reviewer #3 (Remarks to the Author):

REVIEWER COMMENTS

Reviewer #1 (Remarks to the Author):

Thank you for providing us with the opportunity to review the responses of the authors.

We are pleased to see that the authors have comprehensively addressed our concerns and suggestions in their revised manuscript, especially related to the low biomass samples, their quantification, and decontamination procedures and decision making regarding ‘cleaning of’ the sequence data. It has been helpful to provide easy linkage to (a summary of) the community profiles published previously, as in our opinion, the resistome cannot be interpreted without knowledge on the microbiome composition.

We firmly believe that these revisions enhance the paper's overall impact and credibility. Nevertheless, we still have a few minor points we feel may need to be addressed:

- In supplementary Figure 5, it is challenging to distinguish what represents the antibiotic group versus the naïve group. Perhaps the authors could consider labeling the group names above the figures instead of relying solely on colors.

Response: Thank you for the comment. We have now added the labels of group names to easily distinguish between the antibiotic group versus the naïve group above the figure rather than annotating it based on very similar colors (Supplementary Fig. 5).

- At line 121-122 and 169-170, the authors mention that the resistome in the antibiotic-treated group is more heterogeneous compared to the naïve group. We suggest that the authors back this claim with statistical analysis, such as comparing the beta diversity indices (pairwise distances) within the antibiotic-treated group at different time points (e.g., T1 samples compared to T2, etc.) versus within the naïve group.

Response: We have now created a new figure (Supplementary Fig. 10) to compare the beta diversity indices (pairwise Aitchison distance and Bray Curtis distance) within the early antibiotic-treated and naïve group at time point T1 and corresponding statistical significance is also calculated to strengthen our findings.

- There is inconsistency in the numbers for the prenatal + early group: at line 180, this group size is 8, whereas at line 215, it is listed as 9.

Response: One sample from the prenatal + early treatment group, which lacked identifiable ARGs, was removed from downstream analysis at time point T2. As a result, the cohort size for this group was reduced to 8 as detailed on Line 180. We agree this is confusing. To clarify any potential misunderstanding, we have included additional information in the Supplementary Table 1 regarding the number of nasopharyngeal aspirate samples across the treatment groups (early

antibiotic administration and those not exposed to antibiotics). The table outlines the availability of nasopharyngeal aspirate samples at the time of collection, DNA extraction, sequencing, and during the bioinformatics analyses (Supplementary Table 1)

- Supplementary Figure 7 could benefit from statistical analysis.

Response: Thank you for the comment. We have now done the statistical analysis and added the significance levels of statistical tests on the respective boxplots (Supplementary Fig. 7).

- At line 254, "We noticed that..." could be changed to "We found that..."

Response: Changed as per suggestion (Line: 246).

- In Figure 6, it is unclear what "m2" indicates.

Response: Thank you for noticing it. We have now made this clear in the figure legend (Lines: 872-873). Also, we have now used the full form of m^2 in the main text (Lines: 255, 258-259).

Reviewer #3 (Remarks to the Author):

Response: We are thankful for your co-revision.

REVIEWERS' COMMENTS

Reviewer #1 (Remarks to the Author):

The authors have thoughtfully and thoroughly addressed the reviewers' comments, resulting in substantial improvements to the manuscript.

Reviewer #3 (Remarks to the Author):
